# Selection of a de novo gene that can promote survival of *Escherichia coli* by modulating protein homeostasis pathways

**Idan Frumkin[1] & Michael T. Laub** [1,2]

Cellular novelty can emerge when non-functional loci become functional genes in a process termed de novo gene birth. But how proteins with random amino acid sequences beneficially integrate into existing cellular pathways remains poorly understood. We screened ~$10^8$ genes, generated from random nucleotide sequences and devoid of homology to natural genes, for their ability to rescue growth arrest of *Escherichia coli* cells producing the ribonuclease toxin MazF. We identified ~2,000 genes that could promote growth, probably by reducing transcription from the promoter driving toxin expression. Additionally, one random protein, named Random antitoxin of MazF (RamF), modulated protein homeostasis by interacting with chaperones, leading to MazF proteolysis and a consequent loss of its toxicity. Finally, we demonstrate that random proteins can improve during evolution by identifying beneficial mutations that turned RamF into a more efficient inhibitor. Our work provides a mechanistic basis for how de novo gene birth can produce functional proteins that effectively benefit cells evolving under stress.

A central premise in molecular evolution is that organisms must innovate to survive changing environments. Cellular novelty usually emerges via mutations to existing genes or by mixing-and-matching protein domains[1]. However, evolution may also invent new, functional proteins from scratch, a process termed de novo gene birth[2,3]. Little is known about how often this process occurs and, when it does, how such new proteins provide a benefit to cells[4].

Recent studies have used comparative genomics and synteny-based methods to identify lineage-specific genes that may represent de novo genes[5–9]. However, the designation of lineage-specific genes as de novo genes suffers from high false discovery rates due to homology detection failure[10,11]. For bona fide cases of de novo genes, some biological effects have been reported[12] but whether they have beneficial functions that confer a selective advantage remains unknown in most cases.

How can a given nucleotide sequence become a gene? The 'proto-gene' model for de novo gene birth[13] sets two main requirements: (1) stable expression of a locus and (2) beneficial function of the

emerging gene product. If these conditions are met, natural selection can further improve expression, function and regulation to generate a mature gene integrated into cellular physiology. RNA sequencing and ribosome profiling studies have revealed extensive spurious transcription and translation in species across the tree of life[7,13–17]. These loci could serve as a source for new genes.

A complementary approach to investigating de novo gene birth involves characterizing randomly generated proteins and studying whether they can benefit cells. Natural de novo genes do not necessarily come from purely random sequences because various evolutionary forces shape and bias genomes[18–20]. Nevertheless, finding and characterizing functional proteins with random amino acid sequences can provide a missing rationale for the place of de novo proteins in evolution. Previous work has examined in silico and in vitro properties of such random sequences, including their predicted ability to fold into secondary structures[21], chaperones-assisted solubility[22], ATPase activity[23] and potential affinity for different molecules[24–27].

[1]Department of Biology, Massachusetts Institute of Technology, Cambridge, MA, USA. [2]Howard Hughes Medical Institute, Cambridge, MA, USA.
✉e-mail: laub@mit.edu

However, cellular functions for random genes have rarely been demonstrated in vivo. One recent study reported that random proteins in *Escherichia coli* can have beneficial effects on growth[28] but serious caveats in experimental design were subsequently raised[29,30]. Two recent studies found hydrophobic proteins that provide antibiotic resistance to *E. coli* cells[31,32] by membrane depolarization and stimulation of a membrane-bound histidine kinase. Additional studies identified small random proteins that rescue an *E. coli* auxotroph[33,34], probably by binding to the 5′ untranslated region (UTR) of the *his* operon to increase expression of a compensatory enzyme[33]. Another study found a random protein with an unknown molecular mechanism that promotes survival in high concentrations of copper[35].

Still, the functions that random proteins can assume inside cells remain poorly understood. Here, we screened a library of ~10^8 random genes for their ability to promote growth in the presence of the ribonuclease toxin MazF, finding ~2,000 unique genes that restore growth. Although most function non-specifically to reduce transcription from the promoter driving *mazF*, we found a single random antitoxin of MazF, RamF, that specifically rescues cells from MazF toxicity. We characterized the function of RamF, its specificity for MazF, and the mutational pathways to becoming a more efficient inhibitor. Our experiments indicate that RamF is a well-tolerated cytosolic protein that remodels the physiology of *E. coli* cells by interacting directly with cellular chaperones, leading to MazF proteolysis. Thus, our work demonstrates how a small, random protein can instantly have a beneficial function, integrate into pre-existing cellular pathways and become improved by mutation and selection—thereby revealing a plausible mechanism for de novo gene birth.

## Results

### Selection for functional, random genes that inhibit a toxin

We sought to identify functional and beneficial genes originating from random nucleotide sequences. To this end, we created a library of ~10^8 plasmids, each harbouring a tetracycline-inducible promoter ($P_{tet}$) driving a bicistronic operon with a first open reading frame (ORF) encoding a constant 17-amino acid peptide followed by a second ORF with an ATG start codon and then 50 random NNB codons (Fig. 1a; Methods). This bicistronic design minimizes translation initiation biases due to messenger RNA structures involving the second ORF[36]. Deep sequencing of the initial library demonstrated its high complexity, with 99.42% of the ~370,000 reads being single, unique sequences (Extended Data Fig. 1a). The average length of the random ORFs was 28 amino acids, with 23% of the random genes coding for 51 amino acid proteins (Fig. 1b).

We used this library to select genes that enable cells to grow following induction of the toxin MazF, an endoribonuclease that degrades a range of cellular RNAs to inhibit cell growth[37]. We transformed our library into an *E. coli* strain expressing *mazF* from an arabinose-inducible promoter ($P_{ara}$) on the chromosome. We then induced expression of both the random genes and *mazF* to select those genes that inhibit MazF and promote growth. To enrich for true-positive hits and eliminate case of chromosomal mutations that trivially prevent *mazF* expression (for example, $P_{ara}$ mutations), plasmids from the first round of selection were harvested and used to transform new cells harbouring $P_{ara}$-*mazF* (Fig. 1c).

Deep sequencing of the library after two selection rounds revealed ~2,000 enriched, random genes. We arbitrarily chose five of these genes and tested whether they inhibit two additional toxins: RelE, an unrelated ribonuclease toxin[38], and Hok, a short hydrophobic toxin that depolarizes cell membranes[39]. All five hits could inhibit these toxins, which were also expressed from the arabinose-inducible promoter (Fig. 1d) and failed to inhibit MazF when the toxin was expressed from a vanillate-inducible promoter, $P_{van}$ (Fig. 1d). Thus, these random genes are probably not directly inhibiting toxins and instead preventing transcription from the arabinose promoter. Consistent with this conclusion, we found that three of the random genes reduced the levels of

monomeric, super-folding GFP (msfGFP) expressed from $P_{ara}$ but not from the $P_{van}$ promoter (Fig. 1e).

To identify random genes that inhibit MazF independent of its promoter, we transformed the pool of ~2,000 candidates into an *E. coli* strain in which *mazF* was expressed from $P_{van}$ (Fig. 1c). Two successive rounds of selection for growth on vanillate revealed a single random gene that could inhibit MazF driven by $P_{ara}$ or $P_{van}$ and that did not inhibit RelE or Hok (Fig. 1d). This gene did not affect levels of msfGFP produced from $P_{ara}$ or $P_{van}$ (Fig. 1e). We named this gene *ramF* for random antitoxin of MazF.

### RamF inhibits MazF by inducing its degradation

We sought to understand the molecular function the random protein RamF performs to rescue cells. The gene *ramF* has an ORF of 51 codons and an amino acid composition intermediate between small *E. coli* cytosolic and membrane proteins (Fig. 2a and Extended Data Fig. 2). No proteins with sequence similarity to RamF were found in existing sequence databases. We first replaced the short ORF upstream of *ramF* with a ribosome binding site (RBS) and confirmed the MazF-inhibitory activity of this new gene architecture (Fig. 2b). To confirm that the MazF-inhibitory activity of *ramF* depends on a small protein, rather than RNA, we mutated the start codon and found that this variant of *ramF* failed to inhibit MazF. We also generated a recoded variant of *ramF* with 46 synonymous mutations (representing changes to 30% of nucleotides in the ORF) and found that it could still inhibit MazF when co-expressed. Additionally, the originally selected *ramF* rescued growth inhibition following expression of a synonymously recoded *mazF* (83 mutations, 25% of the ORF) (Fig. 2b). Finally, *ramF* did not inhibit close homologues of the *E. coli* MG1655 *mazF*, the toxin used in our screen, as it did not rescue cells expressing *mazF* from the ECOR27 strain[40] or MG1655 chpB, the closest *mazF* homologue in that strain (Fig. 2c). Together, these findings suggest that *ramF* encodes a new protein that specifically alleviates the toxicity of the MG1655 MazF protein.

We next tested the effects of RamF on MazF levels. We generated a variant of MazF that could be easily used in molecular assays such as immunoblots as it harboured both a C-terminal His$_6$-tag, which does not substantially impact function (Extended Data Fig. 3a) and an E24A substitution, which was shown to reduce but not eliminate, RNase activity[41,42]. Cells producing RamF had lower steady-state levels of MazF(E24A)-His$_6$ compared to cells with an empty vector (EV) (Fig. 2d). Production of MazE, the natural antitoxin of MazF that inhibits its toxicity via direct binding[43], did not reduce MazF levels (Fig. 2d). Producing RamF also reduced the fluorescence of MazF(E24A) fused to msfGFP in individual cells compared to a control strain (Fig. 2e). These observations suggest that RamF inhibits MazF through a different mechanism than MazE, probably by reducing toxin levels. RamF did not reduce levels of ChpB(E24A)-His$_6$ (Fig. 2d), consistent with our finding that RamF did not neutralize ChpB toxicity (Fig. 2c).

Because RamF inhibits MazF in a promoter-independent manner, we hypothesized that RamF increases MazF degradation rather than reducing synthesis. To test this possibility, we treated cells producing MazF(E24A)-His$_6$ with tetracycline to block new protein synthesis and followed MazF(E24A)-His$_6$ levels over time. Cells producing RamF exhibited faster turnover of MazF(E24A)-His$_6$ compared to control cells (Fig. 2F), indicating that RamF rescues MazF toxicity by promoting its degradation.

To identify the protease(s) that degrade MazF, we measured MazF(E24A)-His$_6$ levels in strains producing RamF but lacking each of the major *E. coli* proteases (Fig. 2g). While MazF(E24A)-His$_6$ levels were not substantially changed in Δ*hslV* or Δ*htpX* cells, Δ*clpP* cells showed an increase in MazF(E24A)-His$_6$ levels, suggesting that the ClpP protease helps degrade MazF. Because *ftsH* is essential for viability, we could only examine the effects of Δ*ftsH* in the presence of the *sfhC* mutation[44]. Cells harbouring Δ*ftsH* and the *sfhC* mutation showed substantially elevated levels of MazF(E24A)-His$_6$ compared to an isogenic

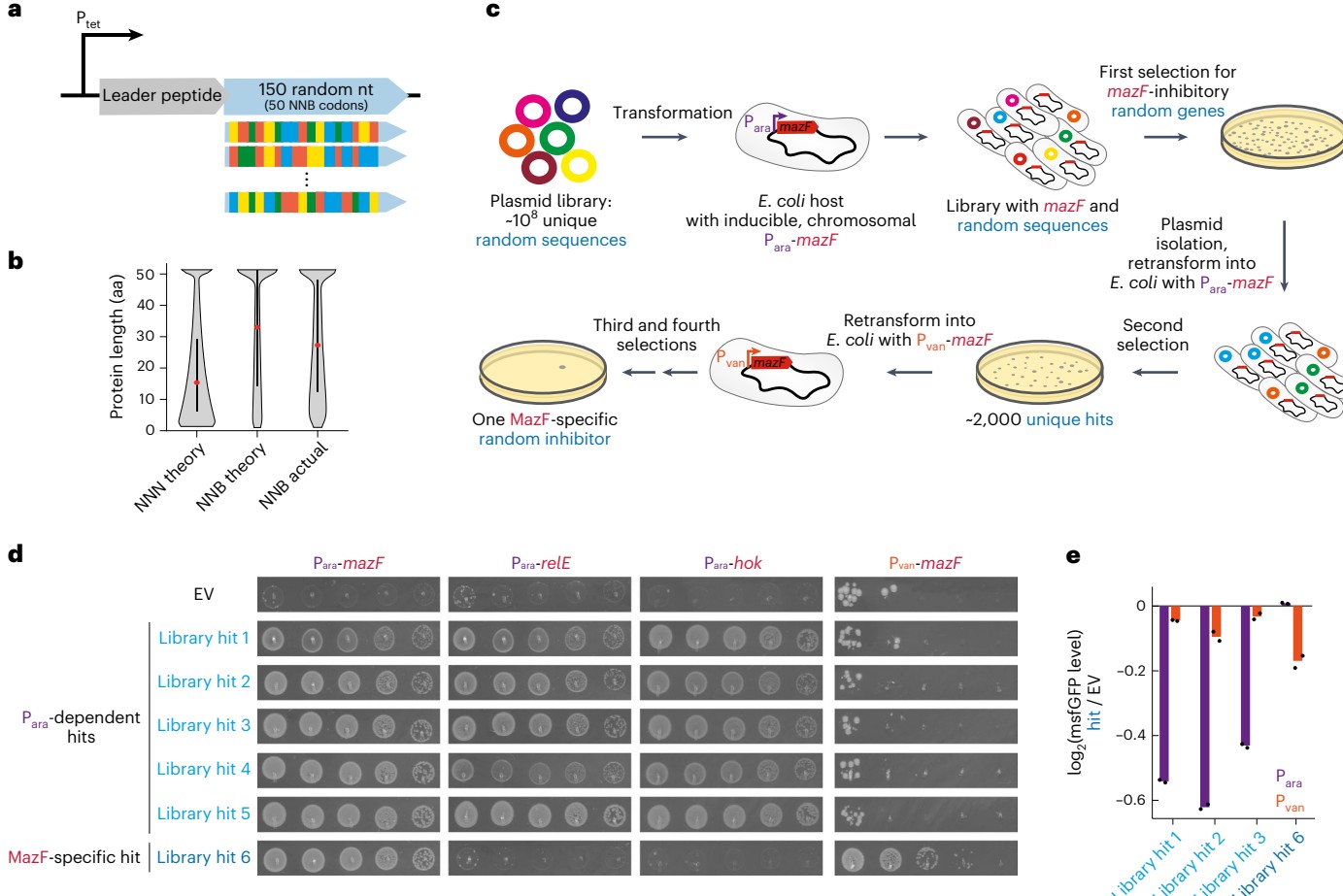

**Fig. 1 | Strategy for selecting functional proteins from a random sequence library. a**, Architecture of the random sequence library. A tetracycline-inducible promoter ($P_{tet}$) drives the expression of a leader peptide followed by an ATG start codon, 150 random nucleotides (50 NNB codons), a stop codon and a transcriptional terminator. **b**, Theoretical protein lengths of 50 NNN codons are lower compared to 50 NNB codons. Actual library distribution as deduced by deep-sequencing preselection is similar to the NNB distribution. Black bar represents 50% of variants and red dot is the median. aa, amino acids. **c**, Selection strategy to identify functional proteins that inhibit the toxin MazF. Approximately $10^8$ plasmids harbouring unique, random genes were transformed into an *E. coli* strain with a chromosomal, arabinose-inducible $P_{ara}$-*mazF* gene. Surviving colonies after *mazF* induction include true hits and false positives due to chromosomal mutations. Plasmids were purified, retransformed

into new cells and selected for a second time. The surviving colonies were then screened twice in a strain expressing *mazF* from the vanillate-inducible promoter ($P_{van}$), resulting in a single gene that passed these selection steps. **d**, Tenfold serial dilution spotting of cells expressing one of the toxins *mazF*, *relE* or *hok* while co-expressing one of the random library hits (numbered 1–6) or an empty vector expressing the leader peptide only. Plasmids harboured the toxins under $P_{ara}$ or $P_{van}$ promoters as indicated. Plasmids carrying the random library hits driven by a $P_{tet}$ promoter. **e**, The $\log_2$ fold-change of median msfGFP fluorescence levels of library hits 1–3 and hit 6 relative to the control strain with an empty vector expressing the leader peptide only. *msfGFP* expressed from either $P_{ara}$ or $P_{van}$, as indicated. Data are the mean of two biological repeats, each black dot is an individual measurement.

---

*sfhC* control, indicating that FtsH plays a key role in MazF degradation. Both Δ*clpP* and Δ*ftsH* strains demonstrated slower degradation rates of MazF(E24A)-His$_6$ compared to control cells when *ramF* was expressed (Extended Data Fig. 4).

We found that MazF(E24A)-His$_6$ levels decreased in the Δ*lon* strain. We first considered whether RamF might inhibit Lon, resulting in increased degradation of MazF(E24A)-His$_6$, thereby phenocopying the Δ*lon* strain. However, RamF did not decrease Lon activity, RamF could inhibit MazF in cells overproducing Lon and producing the known Lon inhibitor PinA did not inhibit MazF (Extended Data Fig. 5a–d). As an alternative, we proposed that RamF might be a Lon substrate such that RamF levels are increased in a Δ*lon* strain, leading to more rapid degradation of MazF(E24A)-His$_6$ in Δ*lon* cells. To test this idea, we created a functional, N-terminally FLAG-tagged RamF (Extended Data Fig. 3b) and compared its steady-state levels in control and Δ*lon* cells. Indeed, FLAG-RamF levels increased in a Δ*lon* strain (Extended Data Fig. 5f).

Because the activity of RamF depends on toxin-induced degradation, we predicted that RamF inhibition efficiency would change in

protease deletion strains that altered MazF levels. Indeed, for Δ*lon* cells in which MazF levels were reduced, RamF was functional at higher MazF induction levels than in control cells (Fig. 2h). In contrast, RamF did not inhibit MazF in Δ*clpP* cells as efficiently as in control cells (Fig. 2h) and it was impossible to transform a plasmid harbouring *mazF* into Δ*ftsH* cells, presumably because even leaky expression leads to enough MazF accumulation and toxicity. As controls, we confirmed that deleting either *hslV* or *htpX*, which did not affect MazF levels, did not affect RamF function. Additionally, we showed that the neutralization of MazF by MazE, which inhibits MazF independent of proteolysis, was not substantially affected by protease deletions.

## RamF interacts with chaperones to modify protein homeostasis

Our results demonstrated that RamF prevents MazF toxicity by facilitating its degradation, particularly via the FtsH protease. Known substrates of FtsH also exhibited decreased steady-state levels in RamF-producing cells (Extended Data Fig. 6a), raising the possibility that RamF activates

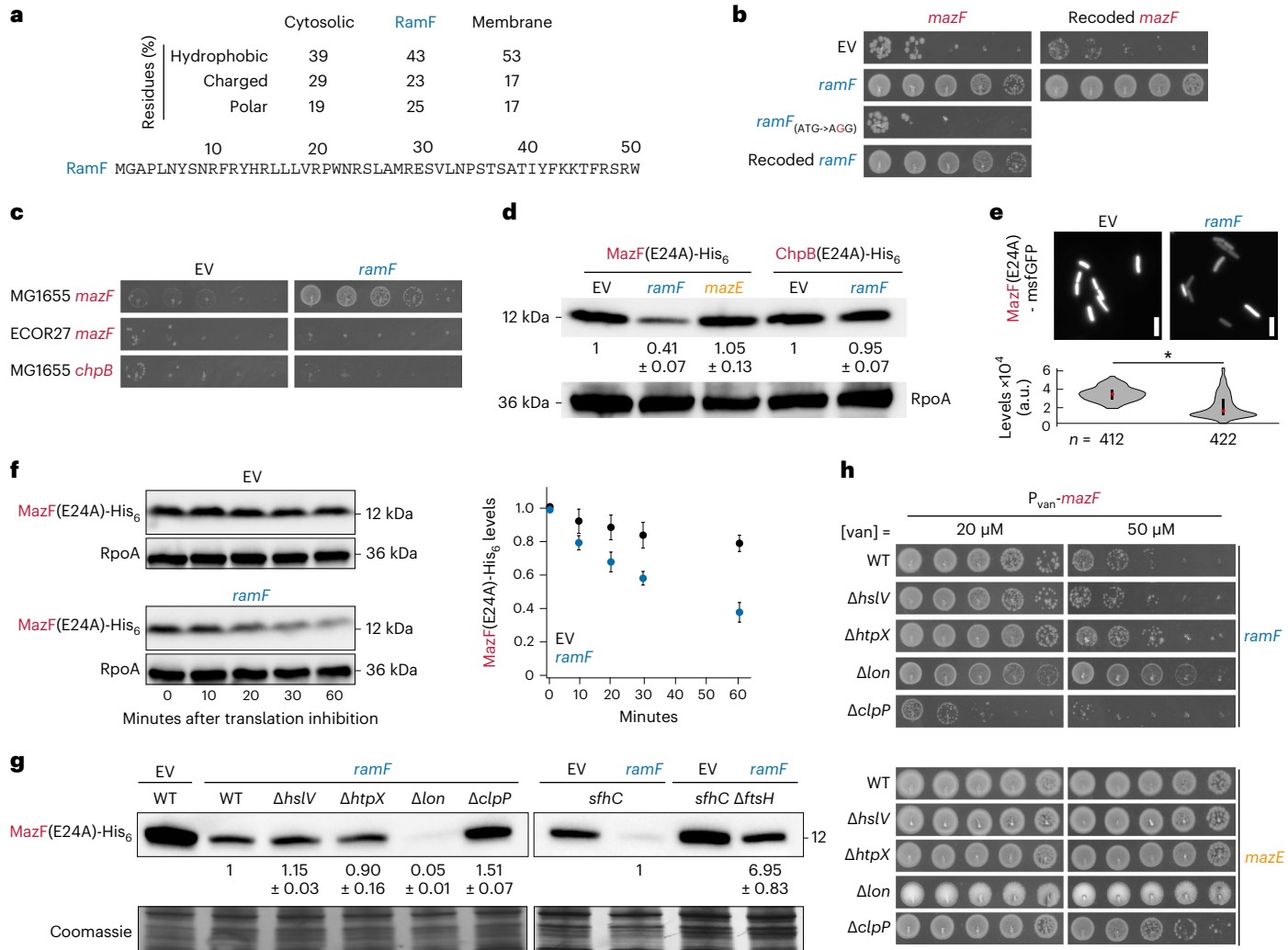

**Fig. 2 | RamF is a specific MazF inhibitor that induces MazF proteolysis.**
**a**, Amino acid sequence of library hit 6, named *ramF* and its amino acid
composition compared to small proteins (<100 amino acids) in *E. coli* that
are either cytosolic (*n* = 181) or membrane-localized (*n* = 80). **b**, Tenfold serial
dilution spotting of cells expressing *mazF* with either its original nucleotide
sequence or a synonymously recoded version from the P_van promoter. Cells
additionally expressed from the P_tet promoter one of the following: *ramF*, *ramF*
with a start codon mutation, synonymously recoded *ramF* or an empty vector.
The leader peptide of the library is not expressed in these cells or experiments
hereafter. **c**, Tenfold serial dilution spotting of cells expressing MG1655 *mazF*
(reference sequence), ECOR27 *mazF* (56% identity) or MG1655 *chpB* (33% identity).
Cells also expressed *ramF* from the P_tet promoter or carried an empty vector.
**d**, Immunoblot of MazF(E24A)-His_6 or ChpB(E24A)-His_6, expressed from P_van, in
cells co-expressing *ramF*, *mazE* or an empty vector. RpoA is a loading control.
Quantification is the mean of *n* = 3 biological repeats and values are normalized
to levels in the empty vector strains. **e**, Fluorescence intensities of MazF(E24A)-GFP

in cells expressing *ramF* or harbouring an empty vector. Violin plots: black bar
represents the middle 50% of cells and red dot is the median. *P = 4.53 × 10^{-93}
based on a two-sided *t*-test, *n* = 412 and 422 cells measured with empty vector
or *ramF*, respectively. Scale bars, 2 μm. **f**, Immunoblot of MazF(E24A)-His_6 from
cells co-expressing *ramF* or harbouring an empty vector. Time points were taken
after the addition of tetracycline to stop the translation of new proteins. RpoA is
a loading control. Quantification is based on the mean of *n* = 3 biological repeats,
error bars represent s.d. and levels are normalized to *t* = 0. **g**, Same as **f** but for
strains also lacking one of the major proteases of *E. coli*, as indicated. Loading
control is based on Coomassie staining of total protein. Quantification for
relevant strains is the mean of *n* = 3 biological repeats and values are normalized
to the *ramF*-expressing strain. Levels in the Δ*ftsH sfhC* strain are normalized to
the *sfhC* control strain. **h**, Tenfold serial dilution spotting of cells co-expressing
*mazF* with either *ramF* or *mazE* in cells also lacking one of the major proteases of
*E. coli*. A plasmid harbouring *mazF* could not be transformed into Δ*ftsH* cells. WT,
wild type.

FtsH. However, overproducing FtsH in cells lacking RamF was insuffi-
cient to inhibit MazF and did not alter RamF efficiency as a MazF inhibi-
tor (Extended Data Fig. 6b), suggesting that RamF does not inhibit MazF
by simply activating FtsH.

How, then, can this random 51 amino acid protein mediate MazF
proteolysis? To characterize the physiological changes caused by
RamF production, we first compared global RNA levels in cells
expressing RamF and an empty vector control. We found that RamF
does not lead to major transcriptional changes (Fig. 3a). There was,
however, an ~2.5-fold upregulation of the native *mazEF* locus (Fig. 3a
left, red dots), supporting a model of RamF-dependent degradation

of MazF because the MazEF complex negatively autoregulates *mazEF*
expression[43,45]; thus, degradation of MazF leads to upregulation
of *mazEF*. In agreement with RamF being a specific MazF inhibitor
(Figs. 1 and 2), the mRNA levels of other toxin–antitoxin (TA) systems,
which are also autoregulated, were not affected (Fig. 3a left, pink
dots, *P* = 0.16, *t*-test).

Because RamF production results in MazF proteolysis, we tested
if the production of RamF affected protein homeostasis pathways,
finding that chaperones and proteases were modestly, but statistically
significantly, upregulated (Fig. 3a, right, *P* = 1.94 × 10^{-4} and *P* = 0.04,
respectively, *t*-test). In comparison, the expression of other gene

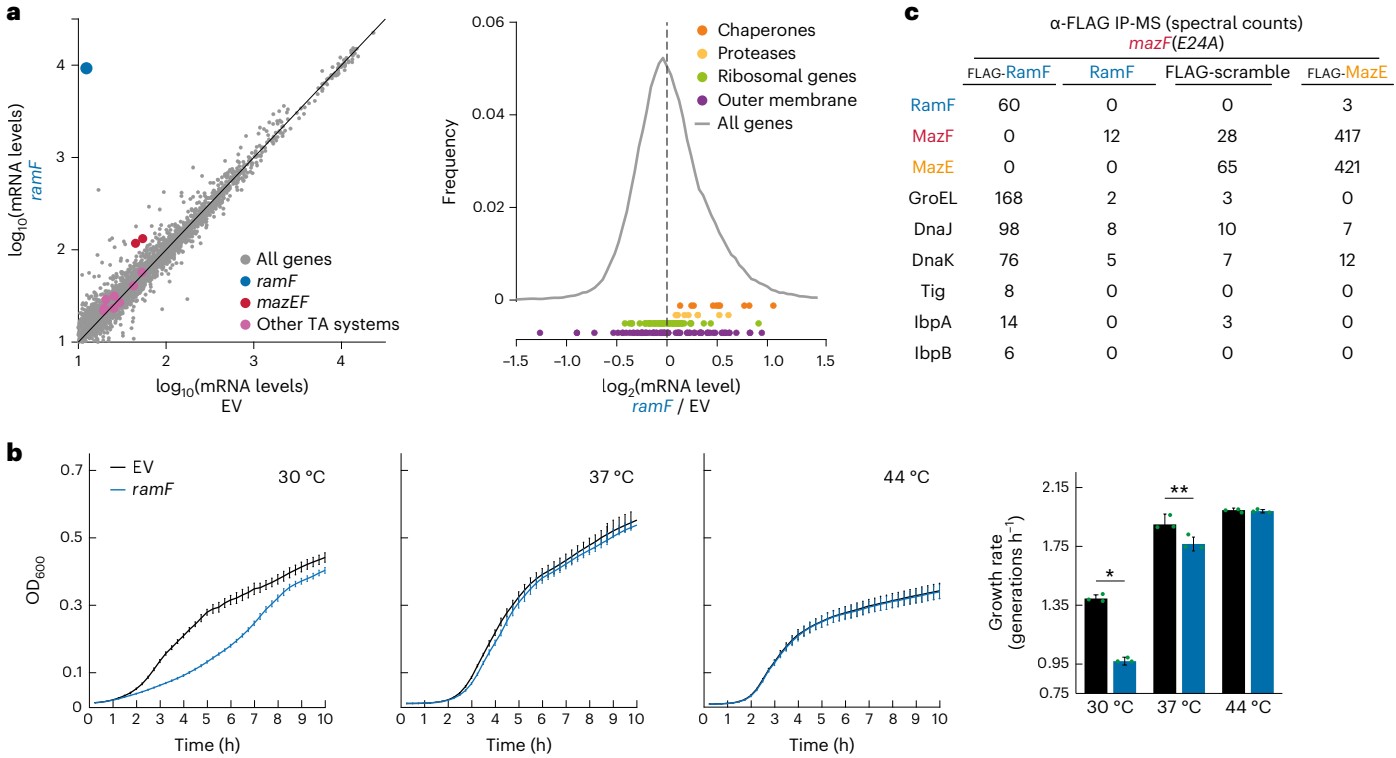

**Fig. 3 | RamF interacts with chaperones. a**, Left, log$_{10}$ of mRNA levels in Transcripts Per Million for *E. coli* genes in cells expressing *ramF* or harbouring an empty vector. Right, log$_2$ of the mRNA level ratio between RamF-producing cells and cells with an empty vector. Colours: grey, all genes; red, *mazEF*; pink, other TA systems; khaki, ribosomal protein genes; purple, outer-membrane genes; orange, chaperones; yellow, proteases. Data based on two biological repeats. **b**, Left, growth curves for cells expressing *ramF* or harbouring an empty vector growing at 30, 37 or 44 °C as a mean of *n* = 3 biological repeats. Right, maximal growth rates calculated from growth curves. \*P = 1.27 × 10$^{-5}$, \*\*P = 0.03 based on a two-sided *t*-test, error bars represent s.d. and each green dot is an individual measurement. **c**, Spectral counts of *E. coli* proteins detected by mass spectrometry following a pull-down with α-FLAG beads from a lysate of cells producing MazF(E24A) and FLAG-RamF, RamF (negative control), FLAG-scrambled-RamF (negative control) or FLAG-MazE (positive control).

groups, for example ribosomal and outer-membrane gene groups, were unaffected (Fig. 3a, right, *P* = 0.41 and *P* = 0.21, respectively, *t*-test).

Our RNA sequencing data suggest that RamF was well tolerated by cells and did not induce a strong stress response. In agreement, producing RamF had a minimal effect (0–2% reduction compared to control cells) on lag times and culture yields at 37 or 44 °C (Extended Data Fig. 7). At 37 °C in LB medium, *ramF* expression led to a small cost in exponential-phase growth rate (Fig. 3b). At 44 °C, *ramF*-expressing cells grew identically to control cells, whereas at 30 °C *ramF* expression caused a severe growth reduction. This temperature-dependent phenotype further indicated that RamF activity may depend on protein homeostasis pathways as chaperone levels are often temperature-dependent[46–52].

To further investigate how RamF affects cell physiology, we sought to find what proteins RamF interacts within cells. We produced functional FLAG-RamF in cells coproducing MazF(E24A), immunoprecipitated RamF using α-FLAG beads and then identified co-eluting proteins by mass spectrometry. We did not detect MazF (Fig. 3c). As a control, we showed that the same procedure using a strain producing FLAG-MazE, did detect MazF, as expected. These results support our conclusion that RamF inhibits MazF via a different mechanism than MazE.

Our mass spectrometry data (Supplementary Table 2) revealed enrichment of multiple proteins that immunoprecipitated with FLAG-RamF but not with two negative control experiments: a strain producing untagged RamF and a strain producing FLAG-tagged but scrambled (same amino acid composition but in a randomized order) RamF protein that could not inhibit MazF (Extended Data Fig. 3b). This analysis revealed that RamF strongly interacts with cellular chaperones, including GroEL (Hsp60), DnaK/J (Hsp70), trigger factor and IbpA/B

(Fig. 3c). RamF also appeared to interact with HldD, PepN and SlyD but deletions of each did not affect the ability of RamF to inhibit MazF through induction of toxin proteolysis (Extended Data Fig. 8).

In sum, our results demonstrated that RamF (1) drives increased proteolysis of MazF, (2) promotes increased expression of chaperones and proteases, (3) interacts in vivo with chaperones and (4) results in a growth defect at a temperature where chaperone expression levels are relatively low. On the basis of these findings, we proposed the following model for MazF inhibition by RamF. In cells lacking RamF, chaperones assist MazF to adopt its native, folded state, which can then cleave RNA and thereby inhibit growth (Fig. 4a, left). In cells producing RamF, chaperones become occupied by RamF such that MazF is unable to fold properly, leaving it susceptible to proteolysis, which allows cellular growth (Fig. 4a, right).

To test this model, we first tested if temperature, which is correlated with chaperone levels, affects the ability of RamF to inhibit MazF. Indeed, we found that MazF failed to inhibit growth at 30 °C even in the absence of RamF, possibly because of insufficient chaperone activity to fold MazF. Also, RamF rescued MazF toxicity at 37 °C but not at 44 °C (Fig. 4b). In agreement, we found that MazF expression levels correlate with growth temperature (Extended Data Fig. 9a). Although consistent with our model, growth temperature affects cell physiology in many ways. Thus, to increase chaperone availability in a more controlled manner, we used a strain producing the heat shock sigma factor (σ$^{32}$) encoded by *rpoH*, which regulates many *E. coli* chaperones. We used an RpoH variant with an I54N substitution that prevents the degradation of this protein and thus maintains its activity[53]. RamF failed to rescue cells producing both MazF and RpoH(I54N) at various temperatures (Fig. 4c and Extended Data Fig. 9b). We also generated

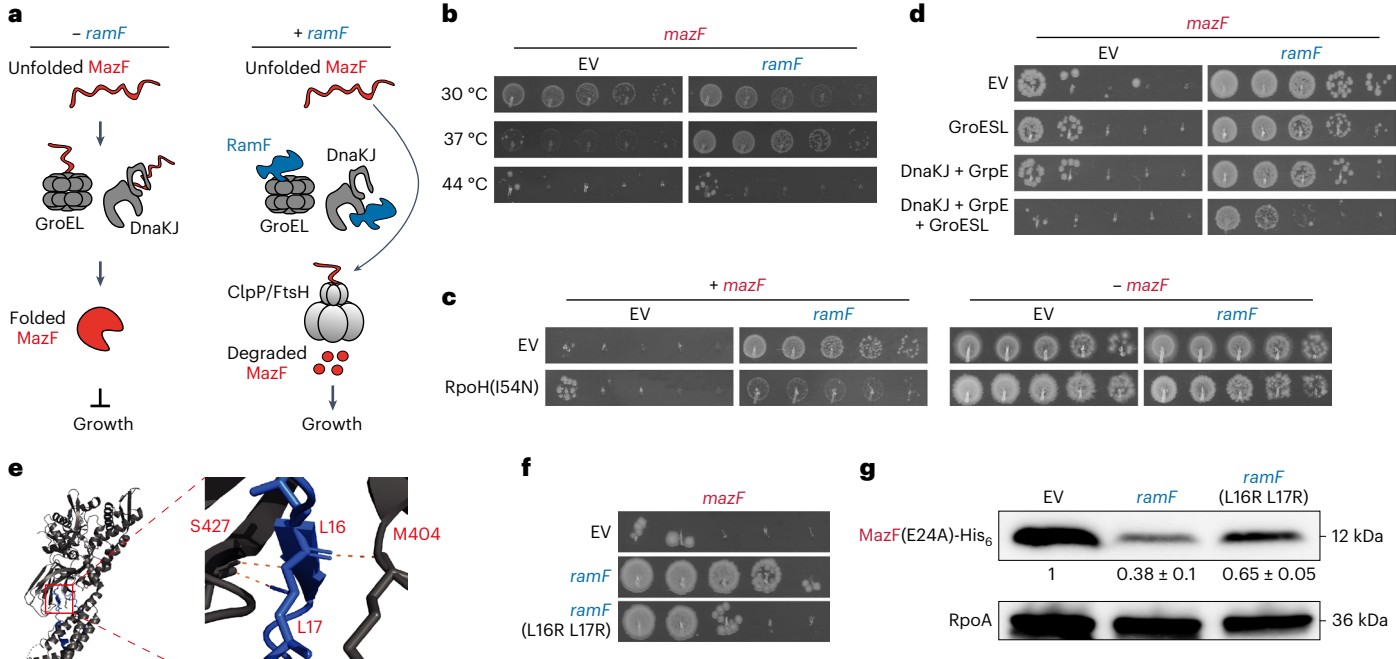

**Fig. 4 | RamF remodels cellular physiology to change protein homeostasis.**
**a**, Model for RamF function as a MazF inhibitor. In cells not producing RamF, chaperones promote proper folding of MazF, leading to widespread RNA degradation and cell growth arrest. In RamF-producing cells, RamF binds chaperones and prevents MazF maturation, allowing FtsH and ClpP to degrade MazF and restore cell growth. **b**, Tenfold serial dilution spotting of cells expressing *mazF* from $P_{ara}$ and co-expressing *ramF* or harbouring an empty vector, incubated at 30, 37 or 44 °C, as indicated. **c**, Tenfold serial dilution spotting of cells expressing *mazF* from $P_{van}$. Cells also express combinations of *ramF*, *rpoH*(I54N) or empty vector, as indicated. **d**, Tenfold serial dilution

spotting of cells expressing *mazF* from $P_{van}$. Cells also express combinations of *ramF*, *groESL*, *dnaKJ + grpE*, *groESL + dnaKJ + grpE* or empty vectors, as indicated. **e**, AlphaFold2 prediction of the interactions between residues M404 and S427 within the substrate-binding domain of DnaK with residues L16 and L17 of RamF. **f**, Tenfold serial dilution spotting of cells expressing *mazF* from $P_{van}$ and co-expressing *ramF*, *ramF*(L16R + L17R) or an empty vector, as indicated. **g**, Immunoblot of MazF(E24A)-His$_6$, expressed from $P_{van}$, from cells co-expressing *ramF*, *ramF*(L16R + L17R) or an empty vector. Loading control is RpoA. Quantification is the mean of $n = 3$ biological repeats and values are normalized to MazF(E24A)-His$_6$ levels in the empty vector strain.

cells that overproduce the chaperone system DnaK/DnaJ/GrpE or GroEL/GroES or both. Overproducing individual chaperone systems partially reduced the ability of RamF to alleviate MazF toxicity, with a substantial drop in RamF activity when overproducing both systems (Fig. 4d). Consistently, overproduction of RpoH(I54N) marginally alleviated the growth defect of RamF-producing cells grown at 30 °C (Extended Data Fig. 9c). Together, these results demonstrate that cellular availability of chaperones is critical to RamF function.

Finally, we asked if the interaction between RamF and chaperones detected in our immunoprecipitation-mass spectrometry (IP-MS) data are important for MazF inhibition. Using AlphaFold2 (refs. 54,55), we modelled the interaction between RamF and DnaK and found that M404 and S427 in DnaK are predicted to bind L16 and L17 in RamF, respectively (Fig. 4e). Notably, these residues in DnaK are found in its substrate-binding domain[56] and were previously shown to bind two contiguous Leu residues of a model peptide[57,58]. A variant of RamF with the substitutions L16R and L17R was not co-immunoprecipitated with DnaKJ as well as the original RamF (Extended Data Fig. 9d). RamF(L16R L17R) also did not inhibit (Fig. 4f) or induce degradation of MazF (Fig. 4g) as efficiently as RamF. Producing RamF(L16R L17R) also resulted in lower overall protein aggregation levels in cells (Extended Data Fig. 9e, see next section). These results are consistent with our model that the interaction of RamF with chaperones is critical to MazF inhibition.

## The N terminus of MazF partially determines RamF specificity

Our results thus far indicate that RamF interacts with central protein homeostasis pathways, which ultimately results in MazF proteolysis.

Using a previously characterized reporter for protein aggregation in *E. coli*[59,60], we found that producing RamF led to increased protein aggregation (Extended Data Fig. 9e), suggesting that the folding of other proteins is affected by RamF chaperone occupancy. Given this function, how does RamF inhibit *E. coli* MG1655 MazF but not other close MazF homologues (Fig. 1b) that share similar predicted structures (Extended Data Fig. 10a)? We speculated that this specificity might stem from *E. coli* MG1655 MazF, but not its homologues, being recognized by FtsH. The FtsH protease can recognize substrates via unique degron sequences at the N or C termini of proteins or internally[61–64]. Because C-terminal tagging of MazF did not change RamF-dependent inhibition (Extended Data Fig. 3b), we tested the relevance of its N terminus to degradation. We fused an N-terminal myc tag to MazF and found that while inhibition by MazE was maintained, the tag abolished inhibition by RamF (Fig. 5a). This result suggests that tagging MazF on its N terminus prevented degradation, presumably by occluding the degron. Indeed, myc-MazF(E24A)-His$_6$ levels did not decrease in cells expressing *ramF* (Fig. 5b). Additionally, removing amino acids 2–6 or 2–10 eliminated MazF toxicity (Fig. 5c), suggesting that this region not only mediates MazF degradation but is essential to MazF toxicity.

A sequence alignment of MG1655 MazF and ECOR27 MazF indicated that the first ten amino acids differ at five positions (Fig. 5d and Extended Data Fig. 10b). We hypothesized that replacing these amino acids in ECOR27 MazF with those of MG1655 MazF might make this chimaeric protein a better FtsH substrate and therefore sensitive to RamF inhibition. Indeed, RamF gained the ability to inhibit ECOR27 MazF when its first ten amino acids matched those in MG1655 MazF (Fig. 5e). Taken together, our results explain how (1) a new, random

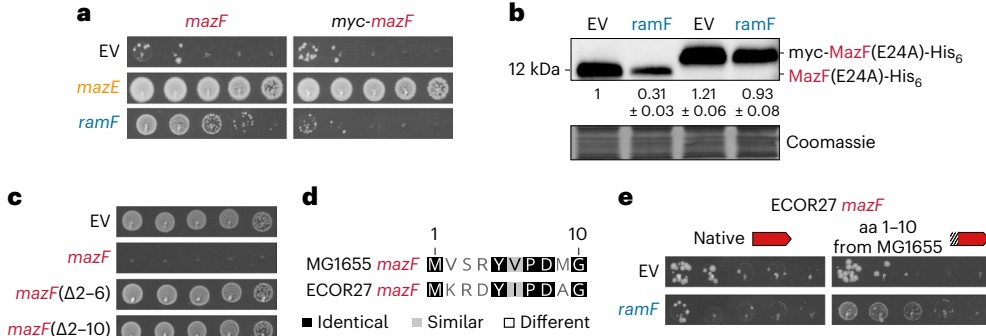

**Fig. 5 | The N terminus of MazF is essential for its inhibition by RamF.**
**a**, Tenfold serial dilution spotting of cells expressing *mazF* or *myc-mazF* from $P_{van}$.
Cells also express *ramF*, *mazE* or an empty vector, as indicated. **b**, Immunoblot
of MazF(E24A)-His$_6$ or myc-MazF(E24A)-His$_6$, expressed from $P_{van}$, from cells
co-expressing *ramF* or harbouring an empty vector. Loading control is based on
Coomassie staining of total protein. Quantification is the mean of $n$ = 3 biological
repeats and values are normalized to MazF(E24A)-His$_6$ levels in the empty vector
strain. **c**, Tenfold serial dilution spotting of cells expressing *mazF*, *mazF*(Δ2–6),
*mazF*(Δ2–10) or an empty vector, as indicated. **d**, Sequence alignment of the
first ten positions of MG1655 MazF and ECOR27 MazF. **e**, Tenfold serial dilution
spotting of cells expressing ECOR27 *mazF* or ECOR27 *mazF*(1–10 from MG1655)
from $P_{ara}$. Cells are additionally expressing *ramF* or an empty vector, as indicated.

protein that interacts with central cellular pathways can have a specific
effect on a single target and (2) how accumulation of mutations on new
targets can make them susceptible to this effect.

## Mutations that improve RamF as a MazF inhibitor are common

Once a de novo gene like *ramF* is established in a genome, natural selec-
tion can, in principle, improve its activity via subsequent beneficial
mutations. To ask whether RamF can become a better MazF inhibitor,
we used PCR-based mutagenesis to create a library of ~60,000 RamF
variants. This library was transformed into the same *E. coli* strain used in
the initial screen and selected on higher levels of MazF such that MazE
rescues growth but the original RamF cannot (Fig. 6a,b; Methods). The
library was deep-sequenced pre- and postselection to find muta-
tions enriched by the selection (Fig. 6c). We found five mutations that
individually improved the inhibition of MazF: F11L, R12M, T40A, I41T
and W51* by RamF (Fig. 6d). Combinations of these mutations mostly
showed additive phenotypes, except for T40A and I41T which exhibited
strong negative epistasis (Fig. 6d). We also generated an improved RamF
variant harbouring F11L, I41T and W51*, which was the most efficient
MazF inhibitor (Fig. 6d). We confirmed that the RamF(F11L I41T W51*)
variant also reduced MazF(E24A)-msfGFP levels further compared to
cells expressing RamF (Fig. 6e).

What mechanisms could underline the beneficial mutations in
RamF? The W51* nonsense mutation replaces the hydrophobic trypto-
phan with a positively charged arginine at the C terminus of RamF, sug-
gesting that this change stabilizes RamF and increases its steady-state
level. Indeed, we observed an ~20% increase in RamF(W51*) levels com-
pared to RamF (Fig. 6f). We also found that a RamF(R50A W51*) variant
showed an ~40% decrease in expression levels and could not inhibit
MazF (Fig. 6f,g), further indicating that RamF levels impact its function.
Finally, we found that a RamF variant with the I41T substitution led to
higher protein aggregation compared to the original RamF (Extended
Data Fig. 9e). The W51* mutation showed a similar, but less pronounced,
increase in aggregation. These results suggest that beneficial muta-
tions that improve RamF functions are common and easily accessible
by natural selection.

## Discussion

There is increasing interest in the discovery and characterization of
small proteins (<50 amino acids) in biological systems[65–67]. Using new
detection methods[68–72], small ORFs are being discovered across the
tree of life, yet their evolutionary origin is enigmatic. The study of ran-
domly generated proteins can support a de novo origin for natural
small proteins by demonstrating how the former assume beneficial
biological functions.

Here, we selected for random proteins that inhibit the toxin MazF.
We identified ~2,000 hits that block MazF in a promoter-dependent
manner probably by reducing expression from $P_{ara}$, although we have
not characterized these hits in depth. Why did we find considerably
more hits targeting the arabinose promoter than MazF itself? A likely
explanation is that the complex arabinose pathway[73–75] simply provides
more opportunities for random proteins to prevent activation of $P_{ara}$.
Additionally, inhibiting the arabinose pathway may be less likely to
perturb essential cellular functions, allowing more solutions to emerge.
Whatever the case, these hits demonstrate that random proteins can
readily adopt beneficial functions inside cells.

We identified one random protein, RamF, that rescued cells in a
promoter-independent manner through interactions with cytosolic
chaperones that remodel the physiology of *E. coli* cells. RamF was our
only promoter-independent hit from a pool of ~$10^8$ sequences. On
one hand, this is surprising given the tendency of random proteins
to include hydrophobic regions[5,21] and bind chaperones in vitro[22].
However, other hydrophobic random proteins in our library may have
suffered from one of the following shortcomings: (1) a fast turno-
ver that prevents functional interactions with cellular components,
(2) a transmembrane domain leading to membrane localization, (3) a
hydrophobic amino acid composition that leads to toxic aggregation or
(4) activation of the stress responses that offset any beneficial change
in cell physiology.

Our results indicated that RamF is specific to MazF, relative to
other toxins. However, RamF did result in increased overall protein
aggregation levels (Extended Data Fig. 9e), suggesting that the fold-
ing of other proteins was affected by the interaction of RamF with
chaperones. RamF did not inhibit close homologues of MazF, probably
because they lack the N-terminal degron in MG1655 MazF. Alternatively,
higher levels of RamF could be required to impact these other toxins,
underscoring the notion that the genomic and cellular context in which
random proteins emerge can affect their functionality. Whatever the
case, to be selected in nature, a de novo gene must cross an expression
threshold that allows its function.

### Fitness effects of de novo proteins

Overproduction of some yeast de novo gene candidates positively
impacted growth[5], demonstrating the benefit these genes can have for
the fitness of microorganisms. However, other studies have found that
random proteins isolated in functional selections can strongly activate
the cellular SOS response[34], reduce cell growth rate[31] or increase growth
lag time and decrease culture yield[32]. RamF resulted in a substantial
fitness cost at 30 °C, a much lower cost at 37 °C (temperature at which
it was selected) and no fitness cost at 44 °C. What does such a cost

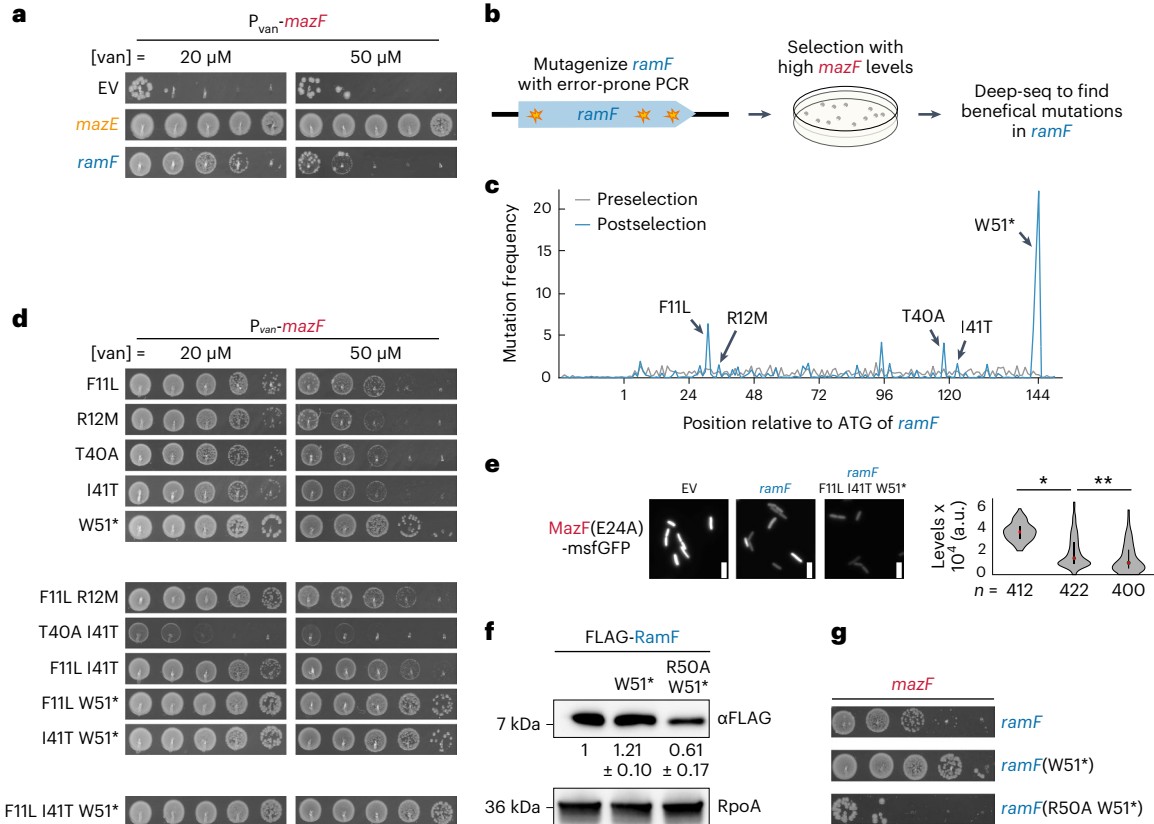

**Fig. 6 | Beneficial mutations that optimize the function of RamF as a MazF inhibitor. a**, Tenfold serial dilution spotting of cells expressing *mazF* from P$_{van}$. Cells are additionally expressing *ramF* or *mazE*, as indicated. **b**, Selection strategy for identifying beneficial mutations that improve RamF activity. Variants of *ramF*, generated by random mutagenesis of *ramF* with error-prone PCR, were selected in the presence of high MazF levels that the original RamF cannot neutralize. **c**, Frequency of *ramF* variants pre- and postselection on high MazF levels. **d**, Tenfold serial dilution spotting of cells expressing *mazF* from P$_{van}$. Cells were additionally expressing *ramF* variants, as indicated. **e**, Fluorescence intensities of MazF(E24A)-GFP in cells expressing *ramF*, *ramF*(F11L I41T W51*) or

harbouring an empty vector. Data for empty vector and *ramF* are as in Fig. 2e. Violin plots: black bar represents the middle 50% of cells and red dot is the median. *$P = 4.53 \times 10^{-93}$, **$P = 9.97 \times 10^{-8}$ based on a two-sided *t*-test, $n = 412$, 422 and 400 cells measured for cells with empty vector, *ramF* or *ramF*(F11L I41T W51*), respectively. Scale bars, 2 μm. **f**, Immunoblot of FLAG-tagged RamF, RamF(W51*) and RamF(R50A W51*) expressed from P$_{tet}$. Loading control is based on RpoA and quantification is the mean of $n = 3$ biological repeats. **g**, Tenfold serial dilution spotting of cells expressing *mazF* from P$_{van}$. Cells were additionally expressing *ramF*, *ramF*(W51*) or *ramF*(R50A W51*).

mean for the chance of a new de novo protein to emerge in nature? A proto-gene probably has a better chance of fixating in an evolving population if producing its protein product does not come with a massive growth cost. Yet, many natural genes have been shown to provide a benefit in some conditions while being deleterious in others[76,77]. Additionally, selection could potentially reduce the costs of a new gene in some conditions through beneficial or compensatory mutations or by ensuring that the gene is only expressed at times it is beneficial.

### Relevance of random proteins to the study of de novo gene birth

We screened a library of random proteins against the toxin MazF but when do biological systems face this challenge? Toxin-antitoxin systems are widespread in bacteria and found on both chromosomes and plasmids[78,79]. Notably, antitoxins for the homologues of a given toxin are often not homologous themselves, suggesting that antitoxins can readily change and possibly arise de novo via a pathway similar to that reported here for RamF. Additionally, antitoxins are often short proteins harbouring unstructured domains, which bind their toxin counterparts[80–84]. Random proteins and young genes also tend to be short and unstructured[21,85,86], further supporting the possibility that some antitoxins have arisen de novo.

　　Although RamF inhibited MazF toxicity, we did not find a random protein that directly interacted with this toxin, like the natural antitoxin

MazE. One explanation could be that more than ~$10^8$ proteins should be screened to find a specific, strong protein–protein interaction and that integration into pre-existing pathways in the cellular system is a more accessible mechanism for random proteins to provide benefits to cells. Such 'physiology modifiers' may be used in cellular evolution as initial but pleiotropic solutions until a more specific one is found. In any case, the idea that the expression of random sequences, probably through spurious transcription and translation, can be advantageous is critical for de novo genes to emerge. Our work demonstrates the feasibility of this randomness-to-function process and provides molecular insight into how de novo genes can integrate into existing cellular pathways.

## Methods

### Plasmids, strains and growth conditions

All strains and plasmids used in this study are listed in Supplementary Table 1. *E. coli* was grown in LB medium (10 g l⁻¹ of NaCl, 10 g l⁻¹ of tryptone, 5 g l⁻¹ of yeast extract) or M9 medium (10× stock made with 64 g l⁻¹ of Na$_2$HPO$_4$·7H$_2$O, 15 g l⁻¹ of KH$_2$PO$_4$, 2.5 g l⁻¹ of NaCl, 5.0 g l⁻¹ of NH$_4$Cl supplemented with 0.1% casamino acids, 0.4% glycerol, 2 mM MgSO$_4$ and 0.1 mM CaCl$_2$). In cases where M9 was used, 0.8% glucose was added to prevent leaky expression from the arabinose-inducible promoter. Media for selection or plasmid maintenance were supplemented with carbenicillin (100 μg ml⁻¹), chloramphenicol (20 μg ml⁻¹) or kanamycin (30 μg ml⁻¹) as appropriate. Overnight cultures were

prepared in the same medium used in a given experiment and cells were grown at 37 °C and 180 rpm in an orbital shaker. The arabinose-, tetracycline- and vanillate-inducible promoters were induced with 0.0002%–0.2% arabinose, 0.1 ng µl$^{-1}$ of anhydrous tetracycline (aTc) and 15–100 µM vanillate, respectively.

Plasmids were generated by Gibson assembly according to the manufacturer's protocol. Inserts were either amplified from a template by PCR or commercially synthesized by Integrated DNA Technology (IDT) as gBlocks. All plasmids were confirmed by Sanger sequencing of the inserts or by full-length plasmid sequencing by Plasmidsaurus. Plasmids were introduced into cells by either TSS transformation or electroporation. DNA and primers used in this study are found in Supplementary Table 3.

### *E. coli* genome engineering
To construct *E. coli* BW27783 *amyA*::P$_{ara}$-*toxin/msfGFP* (strains ML-4045 to ML-4048) and *E. coli* BW27783 *amyA*::P$_{van}$-*toxin/msfGFP* (strains ML-4049 to ML-4050), the 'P$_{ara}$-*toxin, kan$^{R}$*' or 'P$_{van}$-*toxin, kan$^{R}$*' cassettes were PCR amplified from plasmids with primers that included homology to the *amyA* locus. These amplicons were inserted into the genome of the arabinose titratable strain BW27783 (ref. [87]) using the lambda red-based recombination[88]. Single insertions were confirmed by PCR and Sanger sequencing for individual colonies.

### Assembly and transformation of the random gene library
The random gene library was constructed by cloning 150 random nucleotides into the vector ML-4052 such that they immediately followed an ATG and were followed by two TAA stop codons. Specifically, pooled single-stranded DNA oligos of 50 NNB codons flanked on their 5′ end by the sequence GCCTGGCTACCGTCTCGTATG and on their 3′ end by TAATGGAGACGAGCAGGCGATG were synthesized by IDT. To avoid frequent premature stop codons, NNB codons, rather than NNN codons, were used; NNB libraries produce similar amino acid composition to NNN libraries. Oligos were PCR amplified using KAPA enzyme according to manufacturer recommendations with 16 amplification cycles. Six independent reactions were performed and combined to minimize PCR bias. Amplicons of the expected size of 193 nucleotides were purified from a gel using a Zymo Gel DNA Recovery kit and ~500 ng of this insert double-stranded DNA were digested and cut using the type IIS restriction enzyme Esp3I at 37 °C for 3 h to reach full digestion. Approximately 500 ng of the vector ML-4052 were similarly cut by BsmBI and both the insert and vector were subsequently purified on a Zymo DNA clean column. Then, 250 fmol of the vector and 1.25 pmol of the insert were combined in a 20 µl ligation reaction with T4 ligase and Esp3I enzyme. The ligation reaction was cycled between 16 °C for 2 min and 37 °C for 2 min for 100 cycles to allow iterative ligation and digestion. This approach increased the ligation efficiency because once an insert was ligated to a vector it could no longer be cut by the restriction enzyme. Ligations were dialysed on Millipore VSWP 0.025 µm membrane filters for 60 min and then the entire volume was electroporated into 20 µl of Invitrogen MegaX DH10B cells, which resulted in ~10$^8$ transformants. Transformants were grown overnight (14 h) in 50 ml of LB + carbenicillin. Then, the culture was split: 25 ml were frozen in 20% glycerol for long-term storage at −80 °C and 25 ml were prepped for plasmids. The plasmid library of random genes was then dialysed and electroporated into *E. coli* strain ML-4045 to yield ~5 × 10$^8$ transformants.

### Amplicon sequencing of random library and analysis
To assess the library complexity pre- and postselection, random sequences were amplified using a forward primer that included the Illumina anchors and indexes as well as a region directly upstream of the random nucleotides and a reverse primer matching a region immediately downstream of the random nucleotides. PCR reactions were performed using KAPA enzyme according to manufacturer

recommendations with ten amplification cycles. Four independent reactions were performed and combined to minimize PCR bias. Amplicons were purified from an agarose gel using a Zymo Gel DNA Recovery kit. Paired-end sequencing was performed on an Illumina MiSeq at the MIT BioMicro Center. Paired-end reads were merged using PEAR with default parameters and identical reads were clustered using usearch with default parameters.

### Bacterial growth by spotting assay on solid media
In experiments with P$_{ara}$ induction, cultures were grown to saturation overnight in M9-glucose supplemented with 5% LB and the appropriate antibiotics. Cultures were then serially diluted tenfold and spotted on appropriate plates supplemented with 0.8% glucose (toxin repressing), 0.0002%–0.2% arabinose (toxin inducing), 100 ng µl$^{-1}$ of aTc (random gene inducing) or 0.0002%–0.2% arabinose and 100 ng µl$^{-1}$ of aTc (toxin and random gene inducing). Plates were then incubated at 37 °C for 24–36 h before imaging. A similar approach was used in experiments with P$_{van}$ induction, except that LB medium and 15–100 µM vanillate as inducer were used.

### Bacterial growth in liquid
Cultures were grown overnight at 30 °C in an appropriate medium, back-diluted 1:50 and grown an additional overnight at 30 °C. The next day cultures were diluted 1:200 and seeded into a 96-well plate (160 µl culture overlaid with 70 µl of mineral oil) such that each culture had 12 replicates on the same plate and plates were replicated independently at least three times. Growth was monitored at 15 min intervals with orbital shaking on a plate reader (Biotek) at the indicated temperature. Data presented are the mean and standard deviation of all replicates.

### Measurements of msfGFP levels with flow cytometry
Strain ML-4048 or ML-4050 with plasmids ML-4052 to ML-4055 or ML-4058 were grown overnight at 37 °C in LB supplemented with appropriate antibiotics. Cultures were diluted 1:500 in medium supplemented with 100 ng µl$^{-1}$ of aTc to induce expression of the random genes (or an EV) and grown for 30 min at 37 °C. Then, either 0.2% arabinose or 100 µM vanillate was added to induce the expression of msfGFP. Cultures were grown an additional 4.5 h at 37 °C, then diluted 1:40 into PBS supplemented with a high concentration of kanamycin (0.5 g l$^{-1}$) to stop translation and incubated at room temperature for 10 min. Fluorescence was measured on a Miltenyi MACSQuant VYB. Two independent cytometry experiments were performed for each strain and 30,000 cells were measured per replicate. FlowJo was used to analyse the data, gating on single live cells and extracting the median of the msfGFP distribution.

### Western blot analysis of steady-state MazF(E24A)-His$_6$ levels
Cultures were grown overnight at 37 °C in an appropriate medium, back-diluted 1:200 the next day and grown at 37 °C until optical density (OD$_{600}$) ~0.2. Then, 100 ng µl$^{-1}$ of aTc was added to induce *ramF* (or an EV) and cultures were grown for an additional 30 min. When needed, 100 µM vanillate was added to induce *mazF(E24A)-His$_6$* and cultures were grown for an additional 60 min. At OD$_{600}$ ~0.4–0.6, 1 ml of cells was pelleted and flash-frozen. Pellets were then resuspended in 1× Laemmli sample buffer (Bio-Rad) supplemented with β-mercaptoethanol normalized to the OD$_{600}$ of the culture at the moment of collection. Samples were boiled at 95 °C for 10 min, analysed by 4%–20% SDS–polyacrylamide gel electrophoresis and transferred to a 0.2 µm PVDF membrane. To visualize proteins, one of the following primary antibodies was used: (1) anti-His$_6$ (Invitrogen catalogue no. MA1-21315) at a final concentration of 1:1,000, (2) anti-RpoA (Biolegend catalogue no. 663104) at a final concentration of 1:5,000, (3) anti-FLAG (Sigma catalogue no. F1804) at a final concentration of 1:1,000, (4) Anti-DnaK (Abcam catalogue no. ab69617) at a final concentration of 1:1,000 and (5) Anti-DnaJ (Enzo Life Sciences catalogue no. ADI-SPA-410-F) at a final

concentration of 1:1,000. Primary antibodies were incubated overnight at 4 °C. Then, a secondary antibody was added at a final concentration of 1:15,000: (1) goat anti-mouse IgG, HRP (Invitrogen catalogue no. 32430) or (2) goat anti-rabbit IgG, HRP (Invitrogen catalogue no. 32460). SuperSignal West Femto Maximum Sensitivity Substrate (Invitrogen) was used to develop the blots. Blots were imaged by a ChemiDoc Imaging system (Bio-Rad). Images shown are one of at least three independent biological replicates. Band intensities were quantified using ImageJ (https://imagej.nih.gov/ij) and averages and standard errors are based on all replicates. Loading controls were performed using either an anti-RpoA (Biolegend) at a final concentration of 1:5,000 or a Coomassie stain as previously described[89].

## MazF degradation assay
Cultures were grown overnight at 37 °C in an appropriate medium, back-diluted 1:200 the next day and grown at 37 °C until $OD_{600}$ ~0.2. Then, 100 μM vanillate was added to induce *mazF(E24A)-His₆* and cultures were grown for an additional 60 min. Next, 100 ng μl⁻¹ of aTc was added to induce *ramF* (or an EV) and cultures were grown for an additional 30 min. At that point, 1 ml of cells was pelleted and flash-frozen. Then 100 μg ml⁻¹ of tetracycline was added to block protein synthesis and samples were collected at time points 10, 20, 30 and 60 min. Immunoblots for samples were performed as described above, using RpoA as a loading control.

## Immunoprecipitation-mass spectrometry (IP-MS)
*E. coli* strains with plasmids ML-4060, ML-4075, ML-4076 or ML-4078 were grown overnight in LB supplemented with appropriate antibiotics at 37 °C. Overnight cultures were back-diluted 1:200 in 50 ml and grown until $OD_{600}$ ~0.2 at 37 °C. Then, 100 ng μl⁻¹ of aTc was added to induce FLAG-RamF or RamF or FLAG-scrambled RamF or FLAG-MazE and cultures were grown for an additional 30 min. Next, 100 μM vanillate was added to induce MazF(E24A) and cultures were grown for additional 60 min. Cultures were pelleted at 4,000*g* for 10 min at 4 °C, supernatant was removed and cells were resuspended in 900 μl of lysis buffer (B-PER II, ThermoFisher) supplemented with protease inhibitor (Roche), 1 μl ml⁻¹ of Ready-Lyse Lysozyme Solution (Lucigen) and 1 μl ml⁻¹ of benzonase nuclease (Sigma). Samples were incubated at room temperature for 15 min, normalized by $OD_{600}$ and centrifuged at 15,000*g* for 20 min at 4 °C. Next, 850 μl of supernatant were incubated with prewashed anti-FLAG M2 magnetic beads (Sigma) for 1 h at 4 °C with end-over-end rotation after which beads were washed three times with a wash buffer free of detergent (25 mM Tris-HCl, 150 mM NaCl, 1 mM EDTA and 5% glycerol). On-bead reduction, alkylation and digestion were performed. Proteins were reduced with 10 mM dithiothreitol (Sigma) for 1 h at 56 °C and then alkylated with 20 mM iodoacetamide (Sigma) for 1 h at 25 °C in the dark. Proteins were then digested with modified trypsin (Promega) at an enzyme/substrate ratio of 1:50 in 100 mM ammonium bicarbonate, pH 8 at 25 °C overnight. Trypsin activity was halted by the addition of formic acid (99.9%, Sigma) to a final concentration of 5%. Peptides were desalted using Pierce Peptide Desalting Spin Columns (Thermo) and then lyophilized. The tryptic peptides were subjected to liquid chromatography with tandem mass spectrometry. Peptides were separated by reverse-phase high-performance liquid chromatography (Thermo Ultimate 3000) using a Thermo PepMap RSLC C18 column over a 90 min gradient before nano-electrospray using an Exploris mass spectrometer (Thermo). Solvent A was 0.1% formic acid in water and solvent B was 0.1% formic acid in acetonitrile. Detected peptides were mapped to *E. coli* MG1655 protein sequences with the addition of the RamF sequence and protein abundance was estimated by the number of spectrum counts. For full IP-MS results of each pull-down, see Supplementary Table 2.

## RNA extraction and sequencing
*E. coli* strains with plasmids ML-4059 or ML-4060 were grown overnight in LB supplemented with appropriate antibiotics at 37 °C. Overnight

cultures were back-diluted 1:200 in 25 ml of cultures and grown until $OD_{600}$ ~0.2 at 37 °C. Then, 100 ng μl⁻¹ of aTc was added to induce RamF or empty vector and cultures were grown for an additional 45 min. At that time, 1 ml of each culture was mixed with stop solution (110 μl; 95% ethanol and 5% phenol) and pelleted by centrifugation for 30 s at 16,000*g* on a tabletop centrifuge. Pellets were flash-frozen and stored at −80 °C. Cells were lysed by adding TRIzol (Invitrogen) preheated to 65 °C directly to pellets, followed by 10 min of shaking at 65 °C and 2,000 rpm on a ThermoMixer (Eppendorf). RNA was extracted from the TRIzol mixture using Direct-zol (Zymo) columns according to manufacturer's protocol. Genomic DNA was removed by adding 2 μl of Turbo DNase (Invitrogen) in a 100 μl final volume using the provided buffer and incubating for 30 min at 37 °C. DNase reaction products were cleaned up with a Zymo RNA clean and concentrator kit and eluted in 25 μl of water.

Libraries were generated as described previously[37]. The library generation protocol was a modified version of the paired-end strand-specific dUTP method using random hexamer primers. Ribosomal RNA was removed using a recently developed do-it-yourself *E. coli* rRNA depletion kit, using 2.5 mg of total RNA as input[90]. Paired-end sequencing was performed on an Illumina MiSeq at the MIT BioMicro Center.

Geneious Prime 2022.2.2 was used to map reads to the *E. coli* MG1655 genome (accession no. NC_000913) with default parameters and to calculate transcripts per million (TPM) values for all genes. TPM values of each sample were normalized by the median TPM value of a given sample to make all samples comparable[91]. Data shown are based on two independent repeats for each strain. Raw data can be found with NCBI BioSample accessions SAMN32730695 and SAMN32730696.

## Microscopy
*E. coli* strains with plasmid ML-4093 and additional plasmids ML-4059, ML-4060 or ML-4074 were grown in LB supplemented with appropriate antibiotics overnight at 37 °C. Cultures were diluted 1:200, grown at 37 °C for 30 min, supplemented with 100 ng μl⁻¹ of aTc to induce RamF or empty vector and cells were grown for additional 30 min. Next, 0.2% arabinose was added to induce msfGFP and cells were grown for 2.5 h at 37 °C. Then 1 μl of each culture was spotted onto a 1% agarose pad prepared with PBS and placed in a 35 mm glass-bottom dish with 20 mm microwell no. 0 coverglass (Cellvis). Phase-contrast and epifluorescence images were taken using a Hamamatsu Orca Flash 4.0 camera on a Zeiss Observer Z1 microscope using a ×100/1.4 oil immersion objective and an LED-based Colibri illumination system using MetaMorph software (Molecular Devices). Images were analysed in Fiji using the MicrobeJ plug-in[92]. Individual cells were identified by the phase-contrast image and fluorescence intensity was recorded for each cell, with at least 400 cells for each culture.

## Error-prone PCR mutagenesis of RamF
RamF was mutagenized using error-prone PCR-based mutagenesis, as previously described[93]. The gene *ramF* was amplified using Taq polymerase (NEB) and 0.5 mM $MnCl_2$ was added to the reaction as the mutagenic agent. PCR products were treated with DpnI, column purified and cloned into plasmid ML-4059 using Gibson assembly. Gibson products were transformed into DH5α, yielding ~60,000 colonies that were grown overnight at 37 °C. Overnight culture was prepped to obtain the mutagenized library, which was then electroporated into strain ML-4049 and plated on medium containing 100 ng μl⁻¹ of aTc and 100 μM vanillate to induce toxin and *ramF* variants, respectively. The mutagenized library was deep-sequenced pre- and postselection to identify enriched RamF variants that inhibit MazF at a high induction level. These variants were further validated by constructing new plasmids with single, double or triple mutations on *ramF*.

## Protein structure prediction with AlphaFold2

The predicted structure of the DnaK-RamF complex was generated using AlphaFold2 (refs. 54,55), modelling both proteins as monomers with default parameters (MSA method: mmseqs2, pair mode, unpaired; number of models, 5; maximum recycles, 3).

## Reporting summary

Further information on research design is available in the Nature Portfolio Reporting Summary linked to this article.

## Data availability

High-throughput data generated in this study are available with NCBI BioSample accessions SAMN32730695 and SAMN32730696. Source data are provided with this paper.

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

## Acknowledgements

We thank the MIT BioMicro Center and its staff for their support in sequencing; the MIT Biopolymers and Proteomics Core and its staff for their help in mass spectrometry experiments; D. Ding and C. McClune for help with library construction; K. Gozzi, S. Mendoza, A. Murray, S. Srikant and C. Vassallo for comments on the manuscript; P. DeWeirdt, C. Doering, K. Forsberg, M. Guzzo, M. LeRoux, C. Weisman, T. Zhang and all members of the Laub laboratory for helpful discussions. I.F. was supported by a long-term fellowship (LT000706/2018) from the Human Frontier Science Program. M.T.L. is an Investigator of the Howard Hughes Medical Institute.

## Author contributions

I.F. and M.T.L. conceived the project and wrote the manuscript. I.F. designed and performed all experiments and data analysis. M.T.L. supervised the project.

## Competing interests

The authors declare no competing interests.

## Additional information

**Extended data** is available for this paper at https://doi.org/10.1038/s41559-023-02224-4.

**Correspondence and requests for materials** should be addressed to Michael T. Laub.

**A**

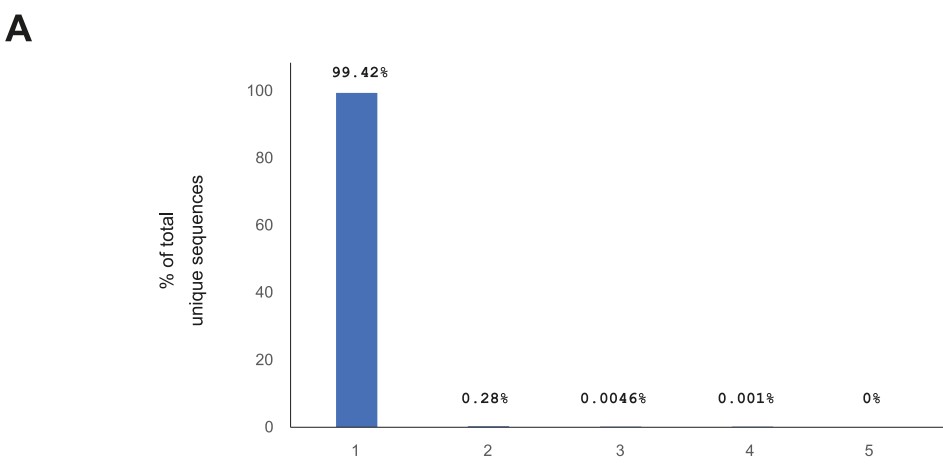

**B**

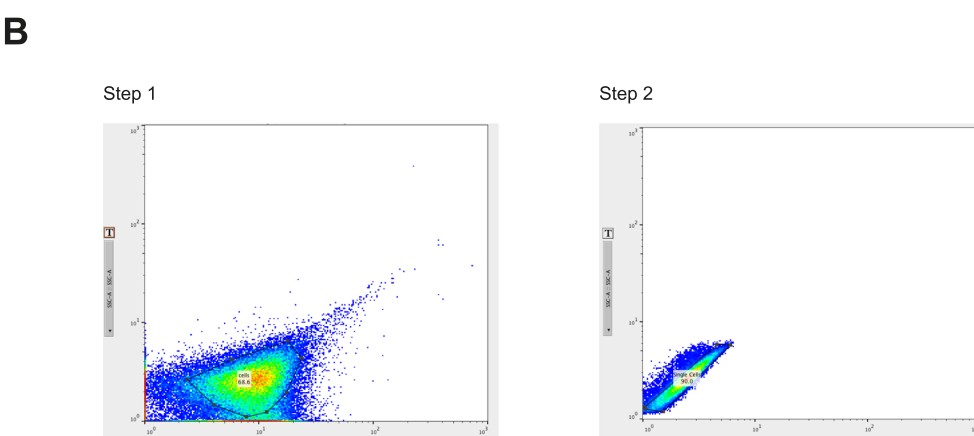

**Extended Data Fig. 1 | Read counts of the library preselection.** (A) Number of reads per unique sequence based on deep sequencing of the random protein library preselection. The total read count was ~300,000. (B) Example for the cytometer gating strategy used throughout this study: First, differentiation between cells and non-cell events using SSC-A and FSC-A parameters. Second, selection of singlet cell events using SSC-A and SSC-H parameters.

**A**

| *E. coli* small proteins (<100 aa) | n= | 181 cytosolic | RamF | 80 membrane |
|---|---|---|---|---|
| % hydrophobic | A | 8.7 | 5.9 | 9.7 |
| | I | 6.0 | 2.0 | 8.4 |
| | L | 9.2 | 11.8 | 13.1 |
| | M | 2.0 | 3.9 | 3.3 |
| | F | 3.2 | 5.9 | 5.5 |
| | W | 1.1 | 3.9 | 2.1 |
| | Y | 2.1 | 5.9 | 2.4 |
| | V | 7.2 | 3.9 | 8.4 |
| % charged | R | 6.4 | 5.7 | 4.4 |
| | H | 2.5 | 2.0 | 1.5 |
| | K | 7.0 | 3.9 | 5.1 |
| | D | 5.5 | 0.0 | 3.3 |
| | E | 8.2 | 2.0 | 3.1 |
| % polar | S | 5.6 | 1.8 | 5.3 |
| | T | 5.2 | 5.9 | 5.3 |
| | N | 4.0 | 7.8 | 3.2 |
| | Q | 4.8 | 0.0 | 3.2 |
| | C | 1.6 | 0.0 | 1.9 |
| | G | 6.0 | 2.0 | 7.1 |
| | P | 3.6 | 5.9 | 3.8 |

**B**

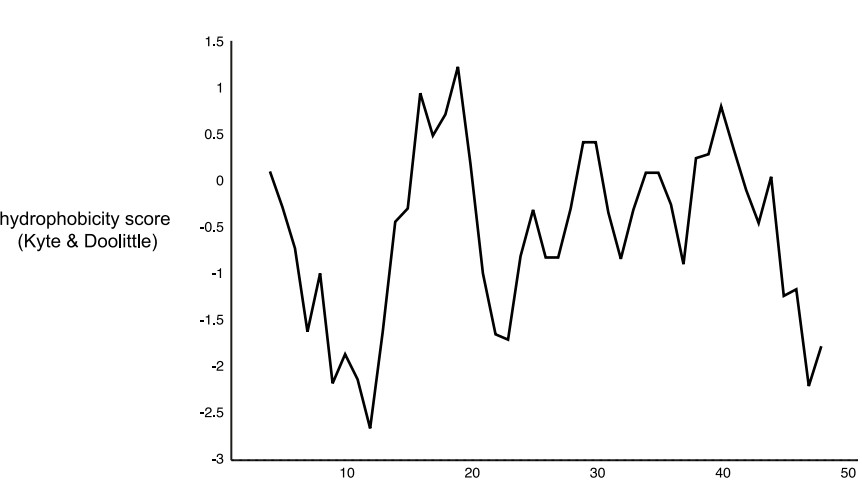

**Extended Data Fig. 2 | Amino acid composition of RamF.** (A) The amino acid composition of RamF compared to cytosolic (n = 181) and membrane (n = 80) proteins in MG1655 *E. coli* whose lengths are each < 100 amino acids. (B) Hydrophobicity plot for RamF based on Kyte & Doolittle scale and an average window size of seven amino acids.

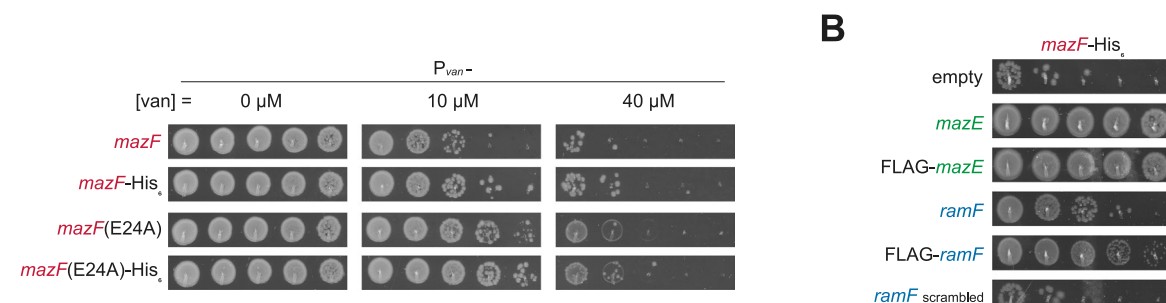

**A**

$P_{van}$-

| [van] = | 0 μM | 10 μM | 40 μM |
|---|---|---|---|
| *mazF* | | | |
| *mazF*-His$_6$ | | | |
| *mazF*(E24A) | | | |
| *mazF*(E24A)-His$_6$ | | | |

**B**

*mazF*-His$_6$

empty
*mazE*
FLAG-*mazE*
*ramF*
FLAG-*ramF*
*ramF* scrambled

**Extended Data Fig. 3 | Epitope-tagging of MazF and RamF does not interfere with their functions.** (A) 10-fold serial dilution spotting of cells expressing *mazF*, *mazF-His$_6$*, *mazF(E24A)*, or *mazF(E24A)-His$_6$*, from $P_{van}$. (B) 10-fold serial dilution spotting of cells expressing *mazF-His$_6$* from $P_{van}$. Cells were additionally expressing *mazE*, *FLAG-mazE*, *ramF*, *FLAG-ramF*, scrambled *ramF*, or an empty vector.

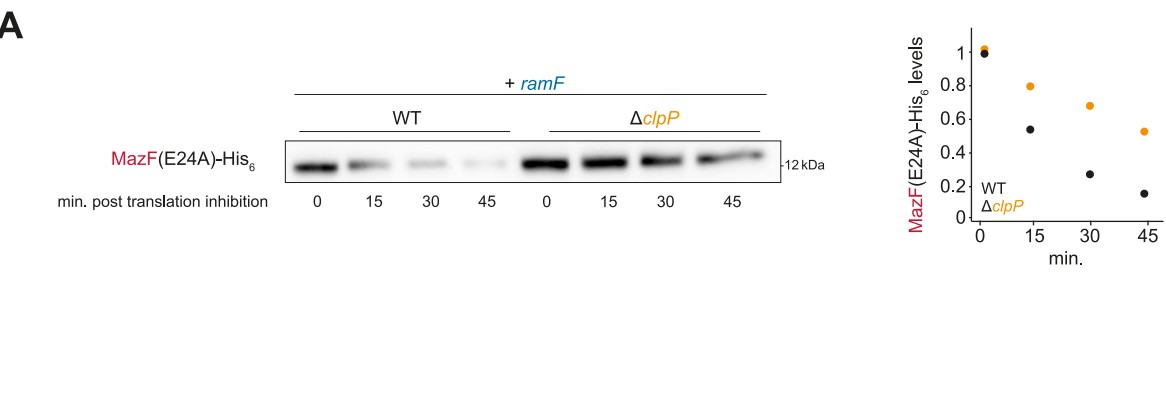

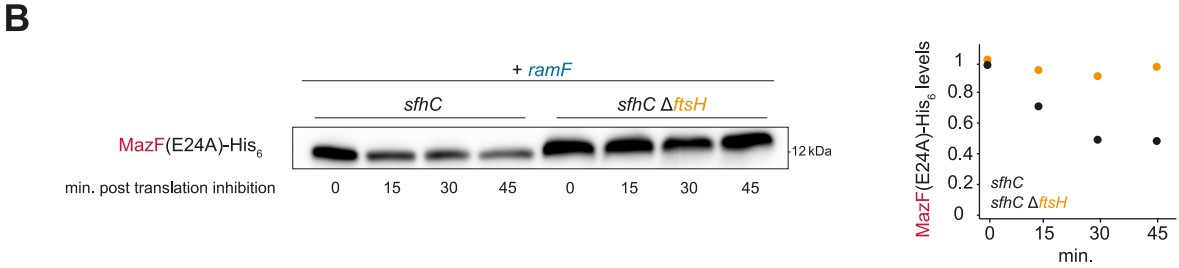

**Extended Data Fig. 4 | MazF(E24A)-His$_6$ proteolytic in Δ$clpP$ and Δ$ftsH$ cells.** Immunoblot of MazF(E24A)-His$_6$, expressed from P$_{van}$, from cells expressing $ramF$ in (A) Δ$clpP$ or (B) Δ$ftsH$ cells. Time points were taken after the addition of tetracycline to stop the translation of new proteins. Quantification is based on two biological repeats and MazF(E24A)-His$_6$ levels are normalized to t = 0.

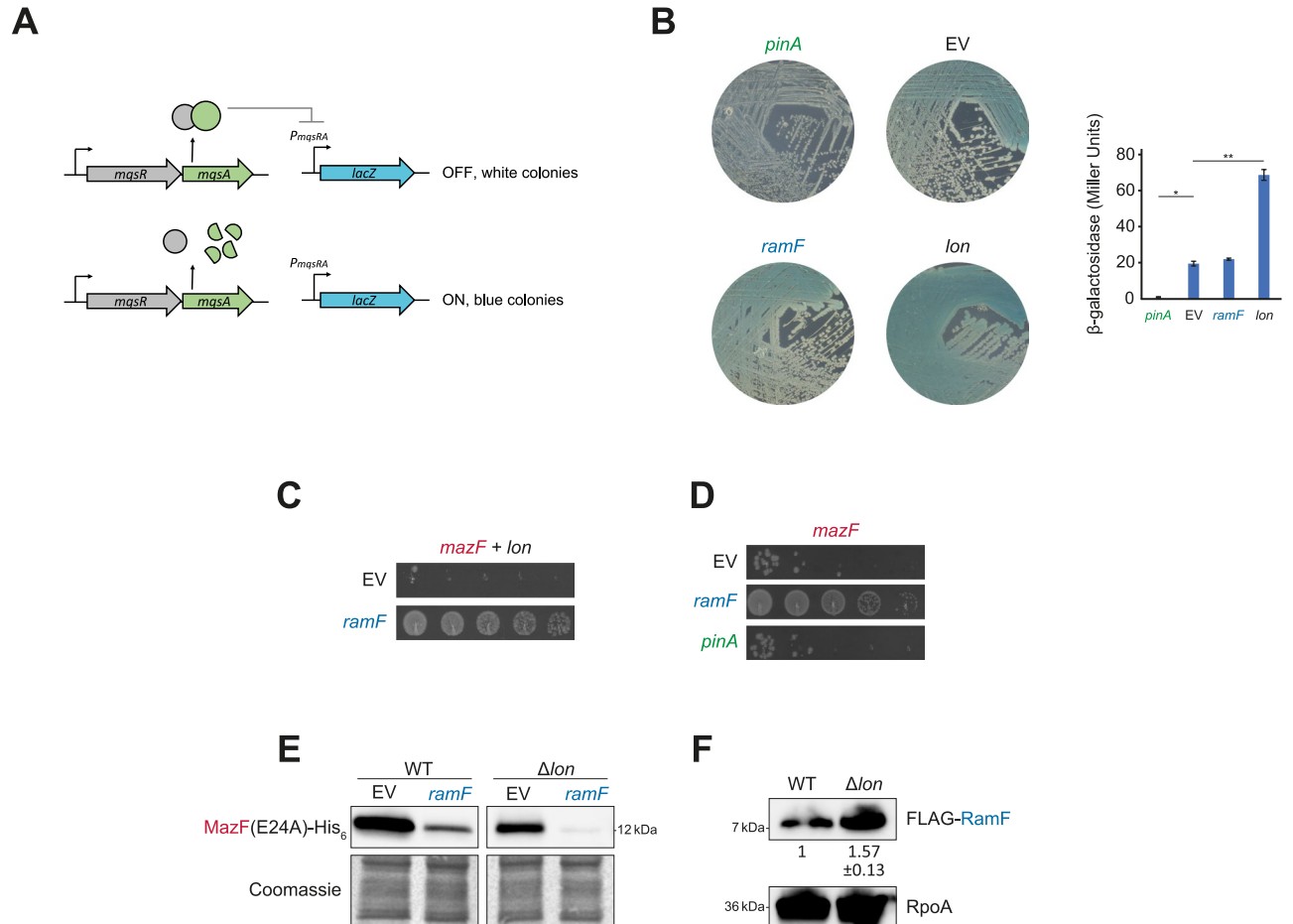

**Extended Data Fig. 5 | RamF does not inhibit the Lon protease.** (A) A system to measure *in vivo* activity of Lon protease: The MqsRA complex inhibits the $P_{mqsRA}$ promoter driving a *lacZ* reporter. In this system, Lon activity is correlated to LacZ production levels because the antitoxin MqsA is a Lon substrate and upon antitoxin degradation, LacZ is produced and colonies turn blue. (B) Cells harbouring the system described in (A) also expressing *lon*, *ramF*, *pinA* (a known Lon inhibitor), or an empty vector. Quantification of β-galactosidase activity in each strain is the based on the mean of n = 3 biological repeats of cells growing at 30 °C overnight. *$P$ = 7.15*10$^{-6}$, **$P$ = 6.36*10$^{-6}$ based on a two-sided t-test and error bars represent SD. (C) 10-fold serial dilution spotting of cells expressing *mazF*,

overexpressing *lon* and additionally expressing *ramF* or an empty vector. (D) 10-fold serial dilution spotting of cells expressing *mazF* and additionally expressing *ramF*, *pinA*, or an empty vector. (E) Immunoblot of MazF(E24A)-His$_6$ expressed from $P_{van}$, in control cells or cells lacking the protease Lon. Cells additionally expressing *ramF* or harbouring an empty vector. Loading control is based on Coomassie staining of total protein. Results represent n = 3 biological repeats. (F) Immunoblot of FLAG-RamF expressed from $P_{tet}$ in control cells or cells lacking the protease Lon. Loading control is based on RpoA and quantification is based on two repeats.

**A**

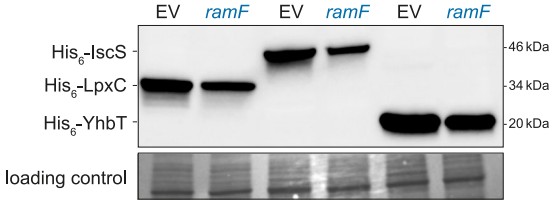

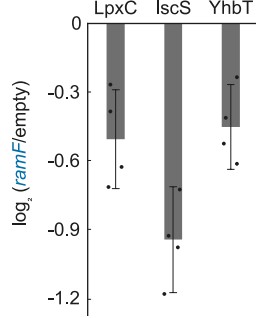

**B**

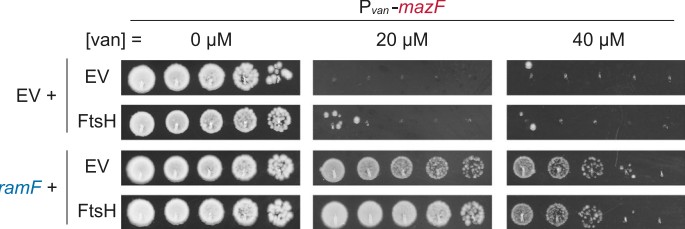

**Extended Data Fig. 6 | Overproduction of FtsH is insufficient to inhibit MazF and does not alter RamF efficency as a MazF inhibitor.** (A) Immunoblot of His$_6$-IscS, His$_6$-LpxC, or His$_6$-YhbT, known FtsH substrates, from cells co-expressing *ramF* or harbouring an empty vector. Loading control is based on Coomassie staining of total protein. Bar: error bars represent SD based n = 4 biological repeats and each black dot is an individual measurement. (B) 10-fold serial dilution spotting of cells co-expressing (i) *mazF*, (ii) empty vector or *ramF* and (iii) empty vector or *ftsH*.

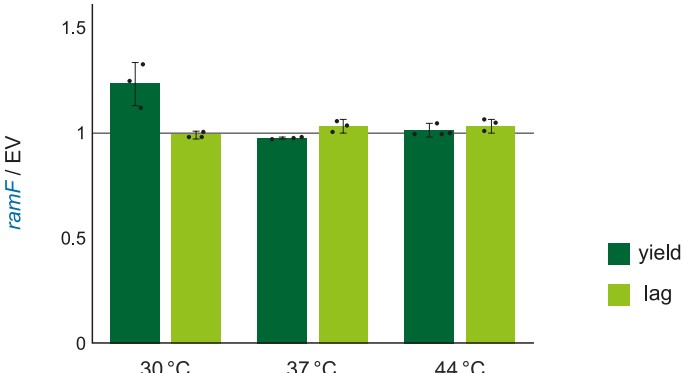

**Extended Data Fig. 7 | Growth characteristics of cells producing RamF.** Lag time (time to reach $OD_{600} = 0.2$) and culture yield (final $OD_{600}$) ratios between cells producing RamF to empty vector at the growth temperatures indicated. Error bars represent SD based on n = 3 biological repeats and each black dot is an individual measurement.

**A**

**B**

**Extended Data Fig. 8 | Non-chaperone proteins that interact with RamF do not affect its ability to inhibit MazF.** (A) 10-fold serial dilution spotting of cells expressing *mazF* in addition to either empty vector or *ramF* in a genetic background of Δ*hldD*, Δ*pepN*, Δ*slyD*, or control cells. (B) Immunoblot of MazF(E24A)-His$_6$ expressed from P$_{van}$, in Δ*hldD*, Δ*pepN*, Δ*slyD*, or control cells. Cells additionally express *ramF* or harbour an empty vector. Loading control is based on RpoA and quantification is based on two repeats.

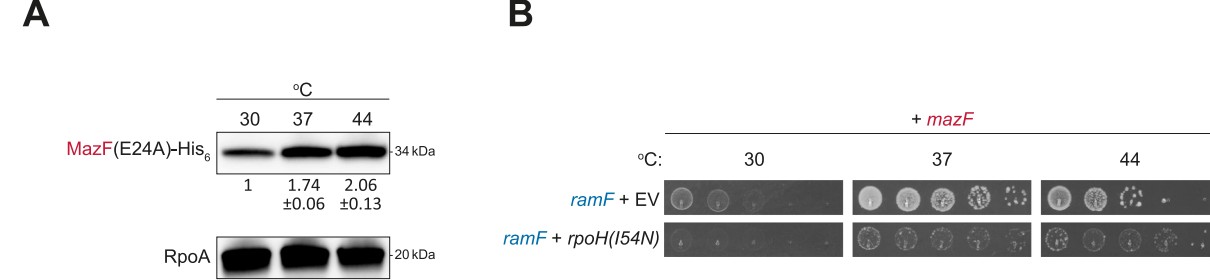

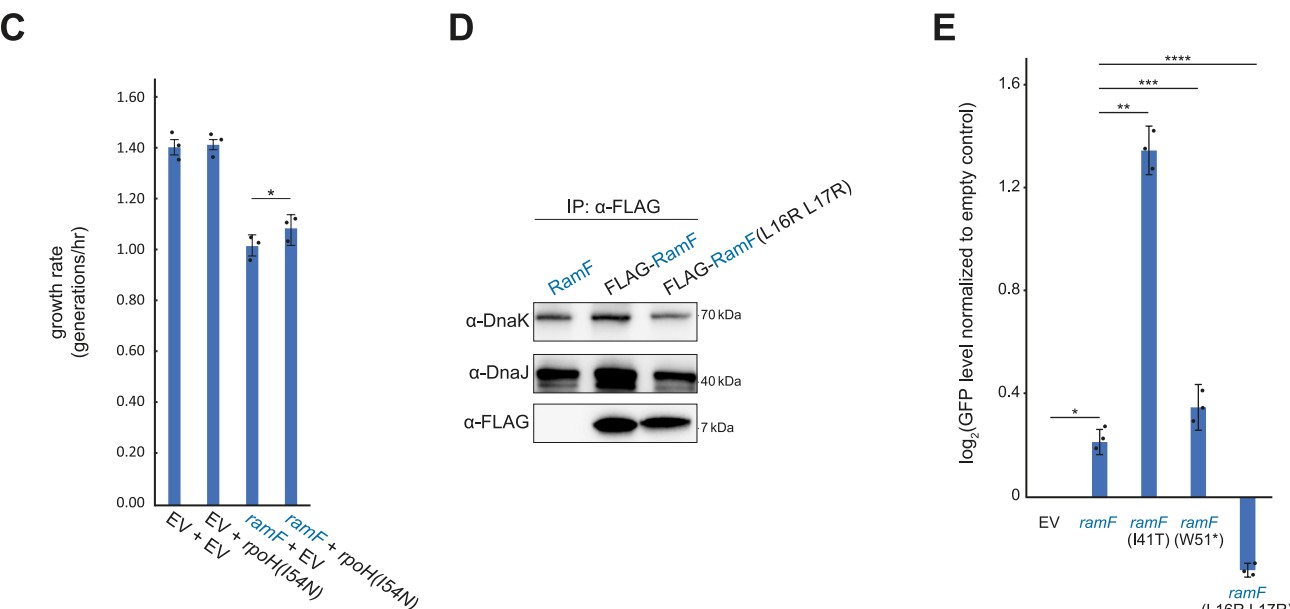

**Extended Data Fig. 9 | MazF and RamF relationship with cellular chaperones.**
(A) MazF(E24A)-His₆ steady-state levels increase with temperature. Immunoblot of MazF(E24A)-His₆ expressed from $P_{van}$, in control cells at growth temperatures 30, 37 and 44 °C. Loading control is based on RpoA and quantification is based on three biological repeats. (B) Overproduction of RpoH(I54N) increases MazF toxicity at a range of temperatures. 10-fold serial dilution spotting of cells expressing *mazF* from $P_{van}$. Cells also express combinations of *ramF*, *rpoH*(I54N), or empty vectors, as indicated and were grown at 30, 37, or 44 °C. (C) Overproduction of RpoH(I54N) alleviates growth rate defect of RamF production. Maximal growth rates (generations per hour) of cells producing combinations of *ramF*, *rpoH*(I54N), or empty vectors, as indicated and grown at 30 °C. Quantification is based on n = 3 biological repeats. *P = 0.04 based on

a one-sided t-test, error bars represent SD and each black dot is an individual measurement. (D) Substitutions L16R and L17R reduce the interaction between RamF and DnaKJ. Cells producing RamF, FLAG-RamF, or FLAG-RamF(L16R L17R) were lysed and used as input for immunoprecipitation using α-FLAG beads. Eluates were then blotted with α-FLAG, α-DnaK and α-DnaJ antibodies. Results represent n = 2 biological repeats. (E) RamF and its variants increase protein aggregation levels. Measurements of msfGFP levels expressed from the $P_{ibpA}$ promoter, whose activity correlates with aggregation levels in *E. coli* cells[53,54]. Cells additionally express *ramF*, *ramF*(I41T), *ramF*(W51*), or *ramF*(L16R L17R). Values are normalized to empty vector control and are based the mean of n = 3 biological repeats. *P = 0.003, **P = 0.00001, ***P = 0.07, ****P = 0.00012, based on a two-sided t-test, error bars represent SD.

**A**

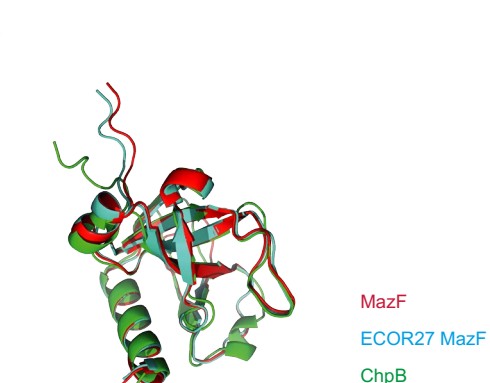

MazF
ECOR27 MazF
ChpB

**B**

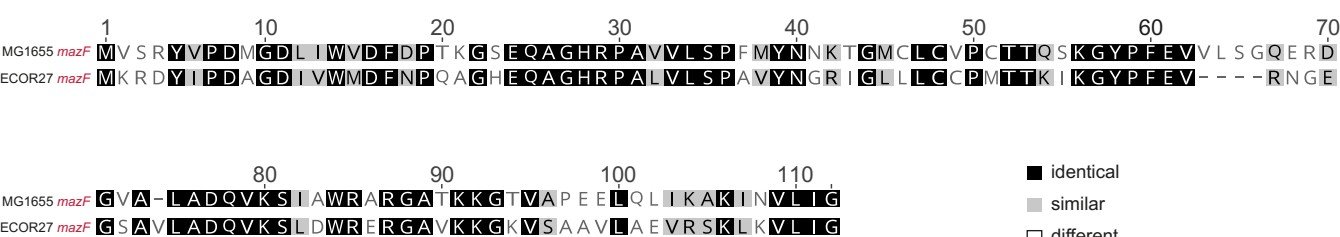

**Extended Data Fig. 10 | AlphaFold2 structure predictions to MazF, ECOR27 MazF and ChpB.** (A) Structures of the three toxins are predicted to be very similar with the very low RMSD scores for structure alignments, as follows: 0.43 for MazF and ECOR27 MazF, 0.76 for MazF and ChpB and 0.64 for ECOR27 MazF and ChpB. Note that the N termini of these three toxins are unstructured and exposed. (B) Amino acid alignment of MG1655 MazF and ECOR27 MazF.

# Reporting Summary

## Statistics

For all statistical analyses, confirm that the following items are present in the figure legend, table legend, main text, or Methods section.

| n/a | Confirmed | |
|---|---|---|
| ☐ | ☒ | The exact sample size (*n*) for each experimental group/condition, given as a discrete number and unit of measurement |
| ☐ | ☒ | A statement on whether measurements were taken from distinct samples or whether the same sample was measured repeatedly |
| ☐ | ☒ | The statistical test(s) used AND whether they are one- or two-sided <br> *Only common tests should be described solely by name; describe more complex techniques in the Methods section.* |
| ☒ | ☐ | A description of all covariates tested |
| ☒ | ☐ | A description of any assumptions or corrections, such as tests of normality and adjustment for multiple comparisons |
| ☐ | ☒ | A full description of the statistical parameters including central tendency (e.g. means) or other basic estimates (e.g. regression coefficient) AND variation (e.g. standard deviation) or associated estimates of uncertainty (e.g. confidence intervals) |
| ☐ | ☒ | For null hypothesis testing, the test statistic (e.g. *F*, *t*, *r*) with confidence intervals, effect sizes, degrees of freedom and *P* value noted <br> *Give P values as exact values whenever suitable.* |
| ☒ | ☐ | For Bayesian analysis, information on the choice of priors and Markov chain Monte Carlo settings |
| ☒ | ☐ | For hierarchical and complex designs, identification of the appropriate level for tests and full reporting of outcomes |
| ☒ | ☐ | Estimates of effect sizes (e.g. Cohen's *d*, Pearson's *r*), indicating how they were calculated |

*Our web collection on statistics for biologists contains articles on many of the points above.*

## Software and code

Policy information about availability of computer code

| Data collection | MetaMorph (v7.10.2.240) (Molecular Devices LLC) was used to collect microscopy data. <br> Biotek Gen5 (v3.02) was used to collect growth curve data. |
|---|---|
| Data analysis | ImageJ (v1.53) and MicrobeJ (v5.13l) were used for image analyses. <br> PEAR (v0.9.11) and USEARCH (v11.0.667) were used for Illumina read clustering. <br> FlowJo (v10.0.7) was used for cytometry date analyses. <br> Geneious Primer (v2022.2.2) was used for RNA-seq analysis. |

For manuscripts utilizing custom algorithms or software that are central to the research but not yet described in published literature, software must be made available to editors and reviewers. We strongly encourage code deposition in a community repository (e.g. GitHub). See the Nature Portfolio guidelines for submitting code & software for further information.

## Data

Policy information about availability of data

All manuscripts must include a data availability statement. This statement should provide the following information, where applicable:
- Accession codes, unique identifiers, or web links for publicly available datasets
- A description of any restrictions on data availability
- For clinical datasets or third party data, please ensure that the statement adheres to our policy

High-throughput data generated in this study is available with NCBI BioSample accessions SAMN32730695 and SAMN32730696.

## Human research participants

Policy information about studies involving human research participants and Sex and Gender in Research.

| | |
|---|---|
| Reporting on sex and gender | N/A |
| Population characteristics | N/A |
| Recruitment | N/A |
| Ethics oversight | N/A |

Note that full information on the approval of the study protocol must also be provided in the manuscript.

# Field-specific reporting

Please select the one below that is the best fit for your research. If you are not sure, read the appropriate sections before making your selection.

☒ Life sciences ☐ Behavioural & social sciences ☐ Ecological, evolutionary & environmental sciences

For a reference copy of the document with all sections, see nature.com/documents/nr-reporting-summary-flat.pdf

# Life sciences study design

All studies must disclose on these points even when the disclosure is negative.

| | |
|---|---|
| Sample size | Sample sizes were chosen based on the number needed to reliably determine differences between groups. All experiments were performed 2-4 times independently. |
| Data exclusions | No data exclusion was performed. |
| Replication | All experimental findings were repeated at least twice. Exact biological repeats are indicated. All reported results were successfully reproduced. |
| Randomization | No experimental groups or control groups were subjectively chosen and there are no covariates to control for as experiments were done in isogenic strains. No experiments required randomization. |
| Blinding | Blinding was not relevant because all data were obtained objectively and had strong effect sizes. |

# Reporting for specific materials, systems and methods

We require information from authors about some types of materials, experimental systems and methods used in many studies. Here, indicate whether each material, system or method listed is relevant to your study. If you are not sure if a list item applies to your research, read the appropriate section before selecting a response.

## Materials & experimental systems

| n/a | Involved in the study |
|-----|----------------------|
| ☐ | ☒ Antibodies |
| ☒ | ☐ Eukaryotic cell lines |
| ☒ | ☐ Palaeontology and archaeology |
| ☒ | ☐ Animals and other organisms |
| ☒ | ☐ Clinical data |
| ☒ | ☐ Dual use research of concern |

## Methods

| n/a | Involved in the study |
|-----|----------------------|
| ☒ | ☐ ChIP-seq |
| ☐ | ☒ Flow cytometry |
| ☒ | ☐ MRI-based neuroimaging |

## Antibodies

**Antibodies used**

6x-His Tag Monoclonal Antibody (clone HIS.H8, Invitrogen Cat#: MA1-21315).
Anti-RpoA antibody (Biolegend Cat#: 663104).
Anti-FLAG antibody (Sigma Cat#: F1804).
Anti-DnaK antibody (Abcam Cat#: ab69617).
Anti-DnaJ antibody (enzo life sciences Cat#:ADI-SPA-410-F).
Goat anti-Mouse IgG Secondary Antibody, HRP (Invitrogen Cat#: 32430).
Goat anti-Rabbit IgG Secondary Antibody, HRP (Invitrogen Cat#: 32460).

**Validation**

All antibodies used in this study are commercial, standard antibodies routinely used in bacterial studies. Manufacturers specify that antibody reactivity is determined by testing in at least one approved application (e.g., western blot). Antibodies were used according to the manufacturer's guidelines. We performed internal validations of antibodies against negative control strains.

## Flow Cytometry

### Plots

Confirm that:

☒ The axis labels state the marker and fluorochrome used (e.g. CD4-FITC).

☒ The axis scales are clearly visible. Include numbers along axes only for bottom left plot of group (a 'group' is an analysis of identical markers).

☒ All plots are contour plots with outliers or pseudocolor plots.

☒ A numerical value for number of cells or percentage (with statistics) is provided.

### Methodology

**Sample preparation**

Strain ML-4048 or ML-4050 with plasmids ML-4052 to ML-4055 or ML-4058 were grown overnight at 37 oC in LB supplemented with appropriate antibiotics. Cultures were diluted 1:500 in medium supplemented with 100 ng/μL aTc to induce expression of the random genes (or an empty vector) and grown for 30 minutes at 37 oC. Then, either 0.2% arabinose or 100 μM vanillate was added to induce the expression of msfGFP. Cultures were grown an additional 4.5 hours at 37 oC, then diluted 1:40 into PBS supplemented with a high concentration of (0.5 g/L) kanamycin to stop translation, and incubated at room temperature for 10 min. Then, fluorescence was measured on a Miltenyi MACSQuant VYB.

**Instrument**

Miltenyi MACSQuant VYB (Miltenyi Biotec)

**Software**

FlowJo

**Cell population abundance**

30,000 cells were measured per replicate and at least 18,000 were left post-gating.

**Gating strategy**

Initial gating was performed using SSC-A and FSC-A and then single bacterial cells were identified using parameters SSC-A and SSC-H.

☒ Tick this box to confirm that a figure exemplifying the gating strategy is provided in the Supplementary Information.

