## [Peer Review File · Nature Ecology & Evolution]

Peer Review Information

Journal: Nature Ecology & Evolution

Manuscript Title: Selection of a de novo gene that can promote survival of E. coli by modulating protein homeostasis pathways

Corresponding author name(s): Michael T. Laub

Editorial Notes:

Reviewer Comments & Decisions:

Decision Letter, initial version:

29th March 2023

Dear Professor Laub,

Your Article, "Selection of a *de novo* gene that can promote survival of *E. coli* by modulating protein homeostasis pathways" has now been seen by 3 reviewers. You will see from their comments copied below that while they find your work of considerable potential interest, they have raised quite substantial concerns that must be addressed. In light of these comments, we cannot accept the manuscript for publication, but would be very interested in considering a revised version that addresses these serious concerns.

We hope you will find the reviewers' comments useful as you decide how to proceed. If you wish to submit a substantially revised manuscript, please bear in mind that we will be reluctant to approach the reviewers again in the absence of major revisions that include new data.

* Include a "Response to reviewers" document detailing, point-by-point, how you addressed each referee comment. If no action was taken to address a point, you must provide a compelling argument. This response will be sent back to the referees along with the revised manuscript.

* If you have not done so already we suggest that you begin to revise your manuscript so that it conforms to our Article format instructions at <http://www.nature.com/natecolevol/info/final-submission>. Refer also to any guidelines provided in this letter.

2[REDACTED]

If you wish to submit a suitably revised manuscript we would hope to receive it within 6 months. If you cannot send it within this time, please let us know. We will be happy to consider your revision so long as nothing similar has been accepted for publication at Nature Ecology & Evolution or published elsewhere.

Nature Ecology & Evolution is committed to improving transparency in authorship. As part of our efforts in this direction, we are now requesting that all authors identified as 'corresponding author' on published papers create and link their Open Researcher and Contributor Identifier (ORCID) with their account on the Manuscript Tracking System (MTS), prior to acceptance. This applies to primary research papers only. ORCID helps the scientific community achieve unambiguous attribution of all scholarly contributions. You can create and link your ORCID from the home page of the MTS by clicking on 'Modify my Springer Nature account'. For more information please visit www.springernature.com/orcid.

Thank you for the opportunity to review your work.

[REDACTED]

Reviewers' comments:

Reviewer #1 (Remarks to the Author):

I very much enjoyed reading the manuscript and I have no major concerns. However, I do have a few random questions/comments that I think may be interesting or important to consider:

1. How "random" are non-coding sequences? On page 3, line 12 you pose the question "how can a random sequence become a gene?". I have no objection to the idea that spuriously transcribed non-genic DNA can acquire a biologically meaningful function and evolve into a new gene. But the use of the word "random" implies that non-genic sequences have random sequences (even if I am quite sure that is not how you mean it). Untranscribed and untranslated sequences very often have some features that are under selection. In bacteria (and phages) the compact intergenic regions are often packed with regulatory sites. Even if a sequence is truly not under selection for any reason it can still evolve due to mutation bias, leading to a non-random sequence composition. I guess what I mean is that I think you should consider replacing the word "random" when you talk about natural sequences,

2but when you describe your library, “random” is absolutely appropriate.

2. In connection to point 1; if you agree that non-genic sequences are non-random, could the sequence composition of extragenic sequences affect the composition or length of possible proto-genes? For example, how does the sequence composition of extragenic sequences in *E. coli* affect the solubility of potentially expressed peptides? Or how often stop codons are encountered? If this makes any sense, could it be worth discussing?

3. Not that it makes much difference now when you have already done all the work, but if you have any reason to continue the screen for more hits or if you plan to do a similar screen in the future: I think you could have reduced the need for the secondary screen (with the Pvan-mazF construct) if the initial screen would have been done with arabinose as sole carbon source (in strains with a functional araBAD operon and the Para-mazF somewhere else on the chromosome). Any peptide that inhibited expression from Para would have prevented growth on arabinose, and only (?) peptides specific for MazF would have been found. This does not affect the current manuscript, but I just wanted to give you the idea anyway.

4. I think even the many hits that appear to be specific towards the Para promoter are interesting and worthy of more characterization. These too are examples of de novo peptides selected from random sequences that show potentially meaningful biological activities. Irrespective of what mechanism(s) they have they show how “easy” it is to find novel peptides that interact with regulatory networks and could potentially evolve into new regulatory mechanisms. Also, depending on how they work and how specific the effect is on the Para promoter, could not some of these peptides be used as an additional level for silencing of the Para promoter in experiments where its inherent leakiness is problematic?

5. In several places in the discussion, you highlight the fact that ramF does not cause any substantial growth defect while other de novo genes isolated by others from random libraries do. Isn't this a bit misrepresenting? Sure, a new gene may have a greater chance of surviving in the long term if it does not cause growth defects, but many genes are known that provide fitness benefits in some conditions but that are deleterious in other conditions. Even the *E. coli* lac operon is a very good example of this: it is absolutely essential for growth on lactose as sole source of carbon and energy but if it is expressed in the absence of lactose, it is deleterious and rapidly inactivated by mutations. Unlike laboratory generated random genes, real de novo genes are unlikely to be highly expressed immediately when they are born. If the main activity of the peptide is toxic under some conditions the gene may need to become integrated in some existing regulatory network before it could reach its full potential. If the toxicity and the beneficial effect of the peptide acts through different mechanisms evolution would probably find a way to improve the beneficial effect while reducing the toxic effect. Perhaps the discussion could be made a little bit more nuanced if you did not press so hard on the growth defects caused by the de novo genes found by others, or if you added a few sentences about how even genes that are deleterious under some conditions could still evolve?

6. I have reservations against the claim that ramF does not cause any substantial growth defect. In figure 3 there is no major effect at 37 degrees but you do not report any growth rates, making it difficult to judge. You account for two more growth parameters (growth yield and lag time) but even these may not be enough to claim that there is no defect (final OD is not necessarily a good estimation of the number of viable cells). Anyway, growth curves and OD measurements are very crude and have a limit of detection for differences of a few percent at best. The gold standard experiment to determine if there are any measurable fitness defects would be a pairwise competition experiment over several growth cycles. Such experiments have the capacity to improve resolution down to the 10⁻³ to 10⁻⁴ range, which is still far above the limit of selection in large populations but

3at least it takes the whole growth cycle into account. If you can not provide data from competition experiments to better support your claim, I would be satisfied if you downplay the claim of no substantial growth defect by RamF expression a bit.

I am looking forward to seeing this manuscript published!

Joakim Näsvall, Uppsala University, Sweden

Reviewer #2 (Remarks to the Author):

In this manuscript, the authors address a highly debated and intensively researched topic of de novo gene birth. They use a library of random genes to select for rescue of MazF-induced growth arrest of *E. coli*. Out of a library of $\sim 10^8$ variants, they select one protein (RamF) that neutralizes MazF toxicity by interaction with chaperones.

The procedure is well described, the number of experiments is astonishing, they are technically sound, and the manuscript is generally very well written. However, in my opinion, some of the conclusions are not fully supported by the data (alternative explanations to the observed phenomenon may exist) and the significance of the study is somewhat overstated. I consider these aspects problematic and do not recommend publication of this manuscript in this form.

To elaborate in more detail, I am listing three major comments:

(1) The effect of RamF on MazF is not direct and virtually disappears when chaperones are overexpressed. As the authors describe, the effect is probably caused mainly by chaperones being occupied by MazF and therefore not available for MazF to fold properly (somewhat in a competition manner) and hence leaving MazF susceptible to proteolysis. In my opinion, this therefore represents an indirect physiological effect and I think this should be a lot more discussed in the manuscript as it represents maybe an extreme case of "function" (it may be even better to call this activity rather than function). Many random/de novo proteins would be expected to impact basic homeostasis and such downstream consequences could be expected. I think that an important experiment (that is missing) would be to perform the pull-down mass spec analysis (presented in Fig. 3C) with 5-10 sequences randomly selected from the $\sim 2,000$ pool of initial hits. This would compare the RamF binding of chaperones with other de novo proteins (albeit with different activity) selected from the random library.

(2) On page 7 of the manuscript, the authors report on the monitoring of MazF levels in strains producing RamF but lacking individually major proteases – to identify which protease degrades MazF. In my opinion, the experiment is not fully conclusive. Although FtsH seems to be the major player, MazF levels are most significantly depleted in the absence of Lon. It is hypothesized that some of the proteases are more active in the absence of Lon (but a reference for such claim is missing). The experiment is missing a control where the effect of the proteases on RamF (a new protein species that could be subject to degradation) would be verified. Since a FLAG-RamF has been used in the project, its stability with the proteases could be monitored. If Lon protease degrades RamF, its quantity in the

4strain lacking Lon could be increased and therefore deplete MazF more extensively. A similar control is also missing in the experiments in Fig S5 – in panel D, cells expressing all mazF, pinA and ramF should be examined.

(3) Similar studies as the one presented in this manuscript have been performed successfully previously by Knopp et al. (on antibiotic resistance), as cited in the discussion. The authors of the referenced studies reported random proteins that provide a specific resistance by (i) membrane depolarization (Knopp et al. 2019) and (ii) by stimulating histidine kinase (Knopp et al. 2021). However, the authors of this manuscript argue that unlike in Knopp et al., RamF does not substantially affect cell growth. They claim that the membrane depolarization protein Arp1 (Knopp et al. 2019) “severely diminishes cell growth both in the presence and absence of antibiotics”, while in the second case (Knopp et al. 2021), it “substantially increased growth lag time and decreased culture yield by 10-20%”.

Unless I overlooked something, this seems to conflict with what Knopp et al. stated in their paper in both cases. To quote:

Knopp et al. 2019 - “Arp1 in *E. coli* causes a similar reduction in fitness, and during exponential growth in nutrient-rich growth medium fitness is reduced by 36%... However, in the presence of aminoglycoside antibiotics it confers a strong fitness advantage, allowing cell proliferation at concentrations which inhibit growth of the wild type. Thus, expression of these peptides offers, similar to SCVs, a conditional benefit to the bacterial cell.”

Knopp et al. 2021 - “The resistance increase was not associated with a visible detrimental growth defect, as all strains retained exponential growth rates similar to the wild type when expressing dcr1 or dcr2 from pRD2 in the presence of 1mM IPTG (S3 Table). Similarly, other growth parameters were also largely unaffected; neither the lag time (defined as time to reach OD600 of 0.01) or final yield (defined as OD600 after 16 hours of growth) decreased more than 10% compared to a wild type when expressed from pRD2 in the presence of 1mM IPTG (S2 Fig).”

In addition, the statement that the cell growth is not substantially affected in this manuscript is (in the light of the evaluation of Knopp et al. study) also not precise. According to Fig. 3B right, the maximal growth rate in ramF expressing cells is affected even at 37C and it is very significant at 30C (as also discussed in the results of the manuscript).

I recommend double-checking the facts and verifying that all the previous studies are cited correctly.

Minor issues:

(1) The legends of the figures should include more detail, especially on the labels of the gels and the controls used. For example, empty vector controls are usually (not always) labeled EV but not specifically explained in the legend.

(2) Some important data are missing statistical tests of significance, e.g. Fig. 3B right.

Reviewer #3 (Remarks to the Author):

5In this study, Frumkin and Laub generated a library of ~108 plasmids capable of expressing genes comprised of ~50 random codons. The authors used this library to select for clones that conferred resistance to the MazF toxin, an endoribonuclease. Not surprisingly, most of the clones that conferred resistance repressed expression from the Para promoter driving the expression of MazF. However, one clone conferred resistance independent of the promoter. Frumkin and Laub further characterized this 51-amino acid random protein that they denoted RamF. Frumkin and Laub report that RamF affect MazF stability by titrating the levels of chaperone proteins required to fold MazF and prevent degradation by proteases. Finally, the authors carried out a screen for optimized RamF derivatives.

This experimental test of how a gene might evolve is very interesting, however we have the following suggestions for improving the study.

1. The authors should expand on some aspects of their study (in some cases carrying out more direct experiments to test conclusions—the descriptions of the results in Figures 2 and 3 at times seem convoluted).

--The very specific effect of RamF on chaperones is quite surprising and raises several questions. How does RamF specifically affect chaperone activity toward MazF but not toward other proteins? How do the levels of MazF and RamF compare to the chaperones? Does DnaKJ overexpression counteract the reduced growth? Given that tagged RamF is much less active, what is evidence that the chaperones that co-purify are not just a consequence of a significant fraction of RamF being misfolded? In addition to answering these questions, the authors should examine RamF effects on other known DnaKJ clients.

--Fig. 2G and Page 7: The authors should examine the RamF-dependent effects on MazF turnover in $\Delta clpP$ and $\Delta sfhC\Delta ftsH$ backgrounds to better support their statement that "RamF prevents MazF toxicity by facilitating its degradation, particularly via the FtsH protease".

--Fig. 3: How are MazF levels affected by growth at different temperatures. How does growth at different temperatures affect the outcome of the experiments shown in panels F and G.

--Fig. 3H-J: Is RamF co-purification with DnaK affected by the L16R+L17R mutation? How is RamF predicted to interact with GroEL?

--Fig. 5 and page 11: The derivation of mutants that are better inhibitors is very interesting. However, this section felt like an add-on. Given that a major focus of this study is how new genes evolve and then are optimized, this contribution would be more substantial if the authors further characterized these optimized RamF derivatives. How do the optimized RamF derivatives alter the proposed interaction with DnaK (Possibly this section could be introduced prior to Fig. 3.)

2. The authors need to describe some results in more detail in the main text:

--Fig. 3A and Page 8: The authors should label and/or provide information about more of the transcripts whose levels change significantly so that the ones chosen for further study do not seem "cherry picked".

6--Fig. 3AC and Page 8: Similarly, the mass spec data set should be more fully described so that the reader can evaluate the specificity of chaperone co-purification (I could not find Table S2).

3. The authors need to further discuss the following:

--Page 6: It was surprising that RamF did not inhibit Mg1655 ChpB or the close homolog from ECOR27. The authors should comment on the similarities/differences in the structures as predicted by AlphaFold as possible explanations for this observation and the conclusion that RamF specificity towards MazF is partially determined by the N terminus of the toxin.

--Page 7: Since ClpP can act with either ClpA or ClpX, the authors need to be more cautious about concluding ClpX involvement based on the phenotypes of the Δ clpP deletion.

--Page 8: Given that ClpP and FtsH both affected MazF levels, why did the authors decide to focus on just FtsH.

--Discussion: How can RamF be a selective modulator of quality control? Presumably the protein is not affecting the substrate binding site. The authors may want to consider restructuring their discussion to end with the big picture rather than effects on the PBAD promoter which were not followed up in the study.

4. More minor comments:

Fig. 2G, 4B. Some loading controls are overexposed. It is not clear why the authors used a Coomassie stained gel as the loading control (was this after transfer?).

Fig. S5B. The differences between the plates are not obvious. The authors could easily carryout a b-galactosidase assay to quantitate the expression.

Author Rebuttal to Initial comments

We thank the reviewers for their enthusiasm about our work and for their constructive feedback for improving it. Below we respond to each query, indicating how the text and/or figures have been modified. Reviewer comments are in black and our responses follow in blue.

Reviewer #1 (Remarks to the Author):

I very much enjoyed reading the manuscript and I have no major concerns. However, I do have a few random questions/comments that I think may be interesting or important to consider:

1. How “random” are non-coding sequences? On page 3, line 12 you pose the question “how can a random sequence become a gene?”. I have no objection to the idea that spuriously transcribed non-genic DNA can acquire a biologically meaningful function and evolve into a new gene. But the use of the word “random” implies that non-genic sequences have random sequences (even if I am quite sure that is not how you mean it). Untranscribed and untranslated sequences very often have some features that are under selection. In bacteria (and phages) the compact intergenic regions are often packed with regulatory sites. Even if a sequence is truly not under selection for any reason it can still evolve due to mutation bias, leading to a non-random sequence composition. I guess what I mean is that I think you should consider replacing the word “random” when you talk about natural sequences, but when you describe your library, “random” is absolutely appropriate.

We agree and have modified this sentence of the Introduction to simply ask “How can a given nucleotide sequence become a gene?” which avoids any labeling of that nucleotide sequence as truly random. We additionally added a sentence clarifying that natural *de novo* genes do not necessarily come from purely random sequences (last paragraph of p. 3).

2. In connection to point 1; if you agree that non-genic sequences are non-random, could the sequence composition of extragenic sequences affect the composition or length of possible proto-genes? For example, how does the sequence composition of extragenic sequences in *E. coli* affect the solubility of potentially expressed peptides? Or how often stop codons are encountered? If this makes any sense, could it be worth discussing?

We agree that this set of questions is interesting and important to address in the future, but it feels somewhat speculative and preliminary to comment on it, particularly without providing analyses of how non-genic DNA in *E. coli* differs from truly random DNA.

3. Not that it makes much difference now when you have already done all the work, but if you have any reason to continue the screen for more hits or if you plan to do a similar screen in the future: I think you could have reduced the need for the secondary screen (with the Pvan-mazF construct) if the initial screen would have been done with arabinose as sole carbon source (in strains with a functional araBAD operon and the

Para-mazF somewhere else on the chromosome). Any peptide that inhibited expression from Para would have prevented growth on arabinose, and only (?) peptides specific for MazF would have been found. This does not affect the current manuscript, but I just wanted to give you the idea anyway.

We appreciate this suggestion and may try it in future screens. The only issue that may arise is that random proteins could possibly prevent expression from P_{ara} enough to prevent toxic levels of MazF from accumulating, without fully compromising growth on arabinose.

4. I think even the many hits that appear to be specific towards the Para promoter are interesting and worthy of more characterization. These too are examples of *de novo* peptides selected from random sequences that show potentially meaningful biological activities. Irrespective of what mechanism(s) they have they show how “easy” it is to find novel peptides that interact with regulatory networks and could potentially evolve into new regulatory mechanisms. Also, depending on how they work and how specific the effect is on the Para promoter, could not some of these peptides be used as an additional level for silencing of the Para promoter in experiments where its inherent leakiness is problematic?

We agree these P_{ara} -specific random genes present an interesting function to explore and their characterization is part of our future plans.

5. In several places in the discussion, you highlight the fact that *ramF* does not cause any substantial growth defect while other *de novo* genes isolated by others from random libraries do. Isn't this a bit misrepresenting? Sure, a new gene may have a greater chance of surviving in the long term if it does not cause growth defects, but many genes are known that provide fitness benefits in some conditions but that are deleterious in other conditions. Even the *E. coli* *lac* operon is a very good example of this: it is absolutely essential for growth on lactose as sole source of carbon and energy but if it is expressed in the absence of lactose, it is deleterious and rapidly inactivated by mutations. Unlike laboratory generated random genes, real *de novo* genes are unlikely to be highly expressed immediately when they are born. If the main activity of the peptide is toxic under some conditions the gene may need to become integrated in some existing regulatory network before it could reach its full potential. If the toxicity and the beneficial effect of the peptide acts through different mechanisms evolution would probably find a way to improve the beneficial effect while reducing the toxic effect. Perhaps the discussion could be made a little bit more nuanced if you did not press so hard on the growth defects caused by the *de novo* genes found by others, or if you added a few sentences about how even genes that are deleterious under some conditions could still evolve?

We thank Dr. Näsval for this comment. We now no longer emphasize the growth defects of the random proteins reported previously (first paragraph of p. 14). Additionally, we have added a paragraph (third paragraph of p. 15) that captures the notion articulated by the reviewer that *de novo* genes may have benefits under some conditions but be detrimental under others.

6. I have reservations against the claim that ramF does not cause any substantial growth defect. In figure 3 there is no major effect at 37 degrees but you do not report any growth rates, making it difficult to judge. You account for two more growth parameters (growth yield and lag time) but even these may not be enough to claim that there is no defect (final OD is not necessarily a good estimation of the number of viable cells). Anyway, growth curves and OD measurements are very crude and have a limit of detection for differences of a few percent at best. The gold standard experiment to determine if there are any measurable fitness defects would be a pairwise competition experiment over several growth cycles. Such experiments have the capacity to improve resolution down to the 10⁻³ to 10⁻⁴ range, which is still far above the limit of selection in large populations but at least it takes the whole growth cycle into account. If you can not provide data from competition experiments to better support your claim, I would be satisfied if you downplay the claim of no substantial growth defect by RamF expression a bit.

We agree and have rephrased our claims about the growth defect of RamF, as described in our response to point 5 above. Additionally, we refer Dr. Näsvalld to Figure 3B in which we provide both growth curves and maximal growth rates at different temperatures. We observed a statistically significant difference of ~6% between cells producing RamF and control cells at 37 °C, which is above the detection level of our system (~1%).

I am looking forward to seeing this manuscript published!
Joakim Näsvalld, Uppsala University, Sweden

Thank you!

Reviewer #2 (Remarks to the Author):

In this manuscript, the authors address a highly debated and intensively researched topic of de novo gene birth. They use a library of random genes to select for rescue of MazF-induced growth arrest of E. coli. Out of a library of ~10⁸ variants, they select one protein (RamF) that neutralizes MazF toxicity by interaction with chaperones. The procedure is well described, the number of experiments is astonishing, they are technically sound, and the manuscript is generally very well written. However, in my opinion, some of the conclusions are not fully supported by the data (alternative explanations to the observed phenomenon may exist) and the significance of the study is somewhat overstated. I consider these aspects problematic and do not recommend publication of this manuscript in this form.

To elaborate in more detail, I am listing three major comments:

(1) The effect of RamF on MazF is not direct and virtually disappears when chaperones are overexpressed. As the authors describe, the effect is probably caused mainly by chaperones being occupied by MazF and therefore not available for MazF to fold properly (somewhat in a competition manner) and hence leaving MazF susceptible to proteolysis. In my opinion, this therefore represents an indirect physiological effect and I

think this should be a lot more discussed in the manuscript as it represents maybe an extreme case of “function” (it may be even better to call this activity rather than function). Many random/de novo proteins would be expected to impact basic homeostasis and such downstream consequences could be expected. I think that an important experiment (that is missing) would be to perform the pull-down mass spec analysis (presented in Fig. 3C) with 5-10 sequences randomly selected from the ~2,000 pool of initial hits. This would compare the RamF binding of chaperones with other de novo proteins (albeit with different activity) selected from the random library.

We respectfully disagree with the reviewer on this point. First, we believe that the term “function”, rather than “activity”, better describes RamF’s ability to inhibit MazF. “Activity” usually refers to enzymes, a class of proteins RamF is not a part of, yet RamF clearly functions as a MazF inhibitor in our experimental system. We therefore think it appropriate to remain with the current descriptor.

Additionally, we find the statement “Many random/de novo proteins would be expected to impact basic homeostasis and such downstream consequences could be expected” to be problematic. It is not at all obvious to us that this would be the expectation and, most importantly, no evidence exists in the literature that random proteins can have *in vivo* beneficial “downstream consequences” because they “impact basic homeostasis”. Indeed, this is a key aspect of our work and thus represents a significant advancement in the field of *de novo* gene birth.

While the reviewer’s interest in the ability of other random proteins to interact with chaperones is understandable, such an experiment would not affect any of our conclusions in this work. A limited IP-MS experiment with ~5 out of 2,000 random proteins having a P_{ara}-specific function may or may not reveal interactions with chaperones. Even if they did, it is already established that they do not influence MazF degradation as RamF does. We would also emphasize that we previously had tested the ability of a scrambled RamF to bind chaperones and found that it does not, indicating that this is not a trivial, universal, and *in vivo* property of random proteins.

We do agree with the reviewer that the ability of RamF to influence MazF via proteostasis networks is a central aspect of our study. This idea was already articulated to some degree in our Discussion, but we have now expanded this section (last paragraph on p. 14) to further note this and to speculate on why RamF is the only protein in our library able to inhibit MazF in this manner.

(2) On page 7 of the manuscript, the authors report on the monitoring of MazF levels in strains producing RamF but lacking individually major proteases – to identify which protease degrades MazF. In my opinion, the experiment is not fully conclusive. Although FtsH seems to be the major player, MazF levels are most significantly depleted in the absence of Lon. It is hypothesized that some of the proteases are more active in the absence of Lon (but a reference for such claim is missing). The experiment is missing a control where the effect of the proteases on RamF (a new protein species that could be subject to degradation) would be verified. Since a FLAG-RamF has been used in the project, its stability with the proteases could be monitored. If Lon protease degrades RamF, its quantity in the strain lacking Lon could be increased and therefore deplete

MazF more extensively. A similar control is also missing in the experiments in Fig S5 – in panel D, cells expressing all *mazF*, *pinA* and *ramF* should be examined.

We have now added Figure S4, showing that the degradation rate of MazF(E24A)-His₆ is slower in both Δ *clpP* and Δ *ftsH* cells compared to control cells when *ramF* was expressed. These data provide additional evidence that these proteases degrade MazF. Following the reviewer's suggestion, we also compared RamF levels between Δ *lon* and control cells and found that RamF levels were indeed increased (Figure S5F). We thank the reviewer for this idea and now discuss the notion that Δ *lon* cells likely show lower levels of MazF because of this increase in RamF levels. These new data are discussed on p. 7 of the revised manuscript.

With regard to Figure S5: as this figure clearly shows that PinA is able to inhibit Lon in our system, we respectfully disagree that measuring its expression levels is a critical control. It is clear that PinA both inhibits Lon and cannot inhibit MazF on its own.

(3) Similar studies as the one presented in this manuscript have been performed successfully previously by Knopp et al. (on antibiotic resistance), as cited in the discussion. The authors of the referenced studies reported random proteins that provide a specific resistance by (i) membrane depolarization (Knopp et al. 2019) and (ii) by stimulating histidine kinase (Knopp et al. 2021). However, the authors of this manuscript argue that unlike in Knopp et al., RamF does not substantially affect cell growth. They claim that the membrane depolarization protein Arp1 (Knopp et al. 2019) “severely diminishes cell growth both in the presence and absence of antibiotics”, while in the second case (Knopp et al. 2021), it “substantially increased growth lag time and decreased culture yield by 10-20%”. Unless I overlooked something, this seems to conflict with what Knopp et al. stated in their paper in both cases. To quote:

Knopp et al. 2019 - “Arp1 in *E. coli* causes a similar reduction in fitness, and during exponential growth in nutrient-rich growth medium fitness is reduced by 36%... However, in the presence of aminoglycoside antibiotics it confers a strong fitness advantage, allowing cell proliferation at concentrations which inhibit growth of the wild type. Thus, expression of these peptides offers, similar to SCVs, a conditional benefit to the bacterial cell.”

Knopp et al. 2021 - “The resistance increase was not associated with a visible detrimental growth defect, as all strains retained exponential growth rates similar to the wild type when expressing *dcr1* or *dcr2* from pRD2 in the presence of 1mM IPTG (S3 Table). Similarly, other growth parameters were also largely unaffected; neither the lag time (defined as time to reach OD600 of 0.01) or final yield (defined as OD600 after 16 hours of growth) decreased more than 10% compared to a wild type when expressed from pRD2 in the presence of 1mM IPTG (S2 Fig).”

In addition, the statement that the cell growth is not substantially affected in this manuscript is (in the light of the evaluation of Knopp et al. study) also not precise. According to Fig. 3B right, the maximal growth rate in *ramF* expressing cells is affected even at 37C and it is very significant at 30C (as also discussed in the results of the manuscript).

I recommend double-checking the facts and verifying that all the previous studies are cited correctly.

Following this reviewer and reviewer 1's suggestions, we have significantly rephrased our claims regarding the fitness cost of random proteins. We no longer emphasize the growth defects of the random proteins reported previously (first paragraph of p. 14). Additionally, we have added a paragraph (third paragraph of p. 15) that captures the notion articulated by reviewer 1 that *de novo* genes may have benefits under some conditions but be detrimental under others.

Minor issues:

(1) The legends of the figures should include more detail, especially on the labels of the gels and the controls used. For example, empty vector controls are usually (not always) labeled EV but not specifically explained in the legend.

We have fixed the specific issue about EV labels and, more generally, tried to ensure that the legends have the necessary information for readers to interpret the figures.

(2) Some important data are missing statistical tests of significance, e.g. Fig. 3B right.

We have fixed this issue in Fig. 3B and throughout.

Reviewer #3 (Remarks to the Author):

In this study, Frumkin and Laub generated a library of ~108 plasmids capable of expressing genes comprised of ~50 random codons. The authors used this library to select for clones that conferred resistance to the MazF toxin, an endoribonuclease. Not surprisingly, most of the clones that conferred resistance repressed expression from the Para promoter driving the expression of MazF. However, one clone conferred resistance independent of the promoter. Frumkin and Laub further characterized this 51-amino acid random protein that they denoted RamF. Frumkin and Laub report that RamF affect MazF stability by titrating the levels of chaperone proteins required to fold MazF and prevent degradation by proteases. Finally, the authors carried out a screen for optimized RamF derivatives.

This experimental test of how a gene might evolve is very interesting, however we have the following suggestions for improving the study.

1. The authors should expand on some aspects of their study (in some cases carrying out more direct experiments to test conclusions—the descriptions of the results in Figures 2 and 3 at times seem convoluted).

--The very specific effect of RamF on chaperones is quite surprising and raises several questions. How does RamF specifically affect chaperone activity toward MazF but not toward other proteins? How do the levels of MazF and RamF compare to the chaperones? Does DnaKJ overexpression counteract the reduced growth? Given that tagged RamF is much less active, what is evidence that the chaperones that co-purify are not just a consequence of a significant fraction of RamF being misfolded? In

addition to answering these questions, the authors should examine RamF effects on other known DnaKJ clients.

Because RamF leads to a ~6% and ~30% growth defect at 37 °C and 30 °C, respectively, we do think that it affects chaperone activity toward other proteins as well, likely leading to their misfolding or aggregation. Because it is hard to guess what proteins will be most affected, we took an alternative approach in which we used a previously developed fluorescent reporter that is sensitive to protein aggregation cellular level. In brief, this reporter fuses the *ibpA* promoter to GFP; this promoter has been shown to be activated following the accumulation of misfolded proteins in *E. coli* (see new refs 53 and 54 in the revised manuscript). Using this reporter, we find (see new Figure S9D) that producing RamF leads to increased aggregation levels compared to cells with an empty vector. We also show that producing RamF variants with mutations that improve its function as a MazF inhibitor leads to a further increase in aggregation levels, as measured by the reporter, and that a defective RamF with mutations that reduce its interaction with DnaK does not lead to activation of the aggregation reporter.

Additionally, we have also now shown (see new Figure S9C) that overproducing RpoH increases the maximal growth rate of RamF-producing cells by ~7% compared to cells carrying an empty vector, consistent with the notion proposed by the reviewer that chaperone overexpression partially mitigates the reduced growth.

With regard to the reviewer's comment that tagged-RamF is much less active, we would note that it is actually the opposite. As shown in Figure S3B, a FLAG-tagged RamF better inhibits MazF compared to the untagged version, suggesting that the IP-MS data reliably reflects RamF's interaction with cellular chaperones. Additionally, our studies of the effect of overproducing chaperones or RpoH on RamF activity (Fig. 3) support the notion that RamF affects chaperone activity and is not simply bound by them in an unfolded state.

--Fig. 2G and Page 7: The authors should examine the RamF-dependent effects on MazF turnover in $\Delta clpP$ and \DeltaftsH backgrounds to better support their statement that "RamF prevents MazF toxicity by facilitating its degradation, particularly via the FtsH protease".

We have now shown (see Figure S4) that the degradation of MazF(E24A)-His₆ is indeed slower in both $\Delta clpP$ and \DeltaftsH cells compared to control cells when *ramF* is expressed.

--Fig. 3: How are MazF levels affected by growth at different temperatures. How does growth at different temperatures affect the outcome of the experiments shown in panels F and G.

We have now shown that MazF levels increase with temperature (see new Figure S9A), and that overproducing RpoH(I54N) increases MazF toxicity at all temperatures when RamF is co-produced (see new Figure S9B). These results support our model of RamF function and the notion that increased chaperone levels promote the folding and stability of MazF.

--Fig. 3H-J: Is RamF co-purification with DnaK affected by the L16R+L17R mutation?
How is RamF predicted to interact with GroEL?

We now show that the L16R and L17R substitutions hamper the ability of RamF to co-purify with DnaK and DnaJ (see new Figure S10), supporting our claim that interactions with chaperones lie at the heart of RamF's function as a MazF inhibitor.

Because GroEL works as an 800-kDa heptamer, AlphaFold was unable to predict its interaction with RamF due to the large size of this complex.

--Fig. 5 and page 11: The derivation of mutants that are better inhibitors is very interesting. However, this section felt like an add-on. Given that a major focus of this study is how new genes evolve and then are optimized, this contribution would be more substantial if the authors further characterized these optimized RamF derivatives. How do the optimized RamF derivatives alter the proposed interaction with DnaK (Possibly this section could be introduced prior to Fig. 3.)

We now show in Figure 5F-G that the W51* mutation results in moderately higher RamF levels, probably due to the change from a hydrophobic amino acid to a positively charged amino acid at the C-terminus. We also now show in Figure S9D that both the I41T and W51* mutations lead to higher aggregation levels compared to the original RamF. These findings suggest that RamF improvement is based on stronger interactions with cellular chaperones to further induce MazF degradation.

2. The authors need to describe some results in more detail in the main text:

--Fig. 3A and Page 8: The authors should label and/or provide information about more of the transcripts whose levels change significantly so that the ones chosen for further study do not seem "cherry picked".

We focused our analysis of the RNA-seq data in Fig. 3A on genes involved in protein homeostasis because of the results in Fig. 2 showing that RamF impacted MazF protein stability. Hence, we wrote (see p. 8 of the manuscript) "Because RamF production results in MazF proteolysis, we tested if the production of RamF affected protein homeostasis pathways, finding that chaperones and proteases were modestly, but significantly, up-regulated (Fig. 3A, right, $P=1.94 \times 10^{-4}$ and $P=0.04$, respectively, t-test)." We think this text conveys that the gene sets examined were not cherry-picked, but rather tested in a hypothesis-driven manner.

We have also now performed a full gene-level analysis, finding that only 36 genes that were up-regulated and 35 genes that were down-regulated in a statistically significant manner. These groups only show biological enrichment for glycerol metabolism (FDR 0.06) and ion transporter (FDR 0.05), respectively, supporting our claims that there are no major transcription changes following RamF production. We have not included this analysis in the main text as we think it does not provide any substantial insight and would bog down the reader. But the full data are, of course, available for interested readers to further examine as they see fit.

--Fig. 3AC and Page 8: Similarly, the mass spec data set should be more fully described so that the reader can evaluate the specificity of chaperone co-purification (I could not find Table S2).

We now describe the mass spectrometry data in more depth (see revised text on p. 9) and include Figure S8 focusing on three non-chaperone proteins that were enriched in our MS data: HldD, PepN, and SlyD. We show that deletion of the genes encoding these proteins did not affect RamF's ability to inhibit MazF through induction of the toxin's proteolysis. We have also ensured that Table S2 is included as Supplementary Material.

3. The authors need to further discuss the following:

--Page 6: It was surprising that RamF did not inhibit Mg1655 ChpB or the close homolog from ECOR27. The authors should comment on the similarities/differences in the structures as predicted by AlphaFold as possible explanations for this observation and the conclusion that RamF specificity towards MazF is partially determined by the N terminus of the toxin.

We now include Figure S11A showing the AlphaFold2 predictions for MazF, ECOR27 MazF, and ChpB with very small RMSD scores for the structure alignments. These structure predictions also suggest that the N-termini of these toxins are unstructured and exposed. We comment on this issue in the first paragraph p. 11 of the revised text.

--Page 7: Since ClpP can act with either ClpA or ClpX, the authors need to be more cautious about concluding ClpX involvement based on the phenotypes of the Δ clpP deletion.

We deleted any reference to ClpA or ClpX and now discuss only ClpP.

--Page 8: Given that ClpP and FtsH both affected MazF levels, why did the authors decide to focus on just FtsH.

We focused on *ftsH* because its deletion results in a stronger effect (~7-fold increase in MazF - see Fig. 2G) compared to a *clpP* deletion (~1.5-fold increase). However, our model in Figure 3D does include both proteases, as both are likely relevant to MazF degradation.

--Discussion: How can RamF be a selective modulator of quality control? Presumably the protein is not affecting the substrate binding site. The authors may want to consider restructuring their discussion to end with the big picture rather than effects on the PBAD promoter which were not followed up in the study.

As explained above in our response to point #1 from this same reviewer, we don't think that RamF's chaperone-binding activity exclusively impacts MazF. Additionally, Figure 4 shows that: (i) tagging the N-terminus of MazF prevents its RamF-induced degradation, and (ii) replacing the first 10 amino acids of ECOR27 MazF with those of MG1655 MazF allows RamF to inhibit this hybrid protein. These findings suggest that these first 10 amino acids of MG1655 MazF are relevant for the recognition of the toxin by proteases and that this degron-like activity is important for the ability of RamF to inhibit MazF. We now comment on this point in the Discussion (second paragraph of p. 15). We have also restructured the Discussion to finish with a big-picture idea instead of the effects with found for the P_{ara} promoter.

4. More minor comments:

Fig. 2G, 4B. Some loading controls are overexposed. It is not clear why the authors used a Coomassie stained gel as the loading control (was this after transfer?).

We have now fixed the loading controls in Fig. 2G, 4B, and S5E and made sure that they are not overexposed. Coomassie staining (as described in PMID 21186791, ref 90 in the text) was done after transfer.

Fig. S5B. The differences between the plates are not obvious. The authors could easily carryout a b-galactosidase assay to quantitate the expression.

Per the reviewer's request, we have now included a quantified β -galactosidase assay in Fig. S5B for this experiment. The conclusions remain the same as initially reported.

Decision Letter, first revision:

15th August 2023

Dear Dr. Laub,

Thank you for submitting your revised manuscript "Selection of a *de novo* gene that can promote survival of *E. coli* by modulating protein homeostasis pathways" (NATECOLEVOL-23020353A). It has now been seen again by the original reviewers and their comments are below. The reviewers find that the paper has improved in revision, and therefore we'll be happy in principle to publish it in Nature Ecology & Evolution, pending revisions to satisfy the reviewers' final requests and to comply with our editorial and formatting guidelines. In particular, we will require some stronger caveats to satisfy Reviewer 2's concerns.

[REDACTED]

Reviewer #1 (Remarks to the Author):

I am happy with the responses and text changes the authors have done to address my comments, but I have a couple of minor remarks:

1. Apparently I missed that the growth rates were reported in the original manuscript. After seeing that there were in fact significant growth rate effects when expressing RamF at 30 and 37 degrees I think the sentence "Although RamF resulted in minimal fitness cost at 37 °C ..." (page 15) is still slightly misrepresenting. Even if a difference in growth rate may appear small, if it is detectable as a change in growth rate it is many orders of magnitude above the limit of selection in bacterial populations. So I don't think you can claim a "minimal fitness cost".

2. I agree with reviewer 2 that the effect of MazF is an activity rather than a function. In my mind, "activity" refers to a biochemical/biophysical process (enzymatic activity, binding activity, inhibitory activity, etc) while "function" refers to a biologically significant activity (i.e. the activity that has evolved and is maintained by selection). For example, enzymes may have promiscuous activities (e.g. acting on non-native or non-natural substrates, or catalysing alternative reactions with their native

18substrate). If such activities are not subject to purifying or positive selection it does not make sense to call them "functions", but they are still significant and measurable activities that could come under selection and allow the enzyme to evolve a new function. Does this make sense?

Reviewer #2 (Remarks to the Author):

I have really enjoyed seeing the additional experiments in the revised version of the manuscript as well as the edits done, mainly in the discussion. I appreciate toning down the claims about the fitness cost of random proteins, relating to previous studies. I value the debate about why only one hit from the tested sequence space affected the MazF levels and how this relates to de novo gene emergence in vivo. I think that all these aspects really improved the manuscript significantly.

While I really appreciate this study, I have to say that I still hold my reservations to the specificity of this effect, which seems to be very sensitive to the actual expression levels of RamF. The initial version of the manuscript already showed that the effect is reverted by chaperone overexpression and the additional experiments (Figure S5F) imply that RamF levels may also be affected throughout the experiments. For now, we know that RamF levels are quite significantly affected by the Lon protease (the only one tested) which makes me wonder whether some effect would also be observed if tested in other protease deletion strains. The differences in MazF levels could then easily be combinations of such effects. This also makes me ask whether the authors have considered this in light of the last paragraph starting on bottom of page 7 (starting by "Because the activity of RamF depends...").

In general, the expression levels of RamF have not been debated while it can have significant effects on the observed phenomena (I think that this issue was also raised by another reviewer). The expression levels of de novo emerging genes are typically very low, and it seems that the phenomenon observed here is quite dependent on the RamF levels. Do the authors consider such effects possible by de novo emerging genes?

Given the number of experiments that have already been done here, I understand the reluctance of the authors to perform additional experiments with the IP-MS. Perhaps only follow-up studies will show how specific/unique the RamF example is. I would argue that exactly because there is so far no in vivo evidence of random protein interaction with chaperones, it is an important experiment to be done (especially since the authors reference a study which implied this interaction in vitro). It would also be great to see some follow up mechanistic/structural studies of the RamF-chaperone interactions (beyond Alpha-fold prediction).

Finally, I believe that the legend of Fig5 F vs G is swapped.

Reviewer #3 (Remarks to the Author):

The authors have made a commendable effort to carefully address all of the reviewers' comment. I think this study is an important contribution.

19Our ref: NATECOLEVOL-23020353A

23rd August 2023

Dear Dr. Laub,

Thank you for your patience as we've prepared the guidelines for final submission of your Nature Ecology & Evolution manuscript, "Selection of a *de novo* gene that can promote survival of *E. coli* by modulating protein homeostasis pathways" (NATECOLEVOL-23020353A). Please carefully follow the step-by-step instructions provided in the attached file, and add a response in each row of the table to indicate the changes that you have made. Please also check and comment on any additional marked-up edits we have proposed within the text. Ensuring that each point is addressed will help to ensure that your revised manuscript can be swiftly handed over to our production team.

****We would like to start working on your revised paper, with all of the requested files and forms, as soon as possible (preferably within two weeks). Please get in contact with us immediately if you anticipate it taking more than two weeks to submit these revised files.****

In recognition of the time and expertise our reviewers provide to Nature Ecology & Evolution's editorial process, we would like to formally acknowledge their contribution to the external peer review of your manuscript entitled "Selection of a *de novo* gene that can promote survival of *E. coli* by modulating protein homeostasis pathways". For those reviewers who give their assent, we will be publishing their names alongside the published article.

20Nature Ecology & Evolution offers a Transparent Peer Review option for new original research manuscripts submitted after December 1st, 2019. As part of this initiative, we encourage our authors to support increased transparency into the peer review process by agreeing to have the reviewer comments, author rebuttal letters, and editorial decision letters published as a Supplementary item. When you submit your final files please clearly state in your cover letter whether or not you would like to participate in this initiative. Please note that failure to state your preference will result in delays in accepting your manuscript for publication.

Cover suggestions

As you prepare your final files we encourage you to consider whether you have any images or illustrations that may be appropriate for use on the cover of Nature Ecology & Evolution.

Nature Ecology & Evolution has now transitioned to a unified Rights Collection system which will allow our Author Services team to quickly and easily collect the rights and permissions required to publish your work. Approximately 10 days after your paper is formally accepted, you will receive an email in providing you with a link to complete the grant of rights. If your paper is eligible for Open Access, our Author Services team will also be in touch regarding any additional information that may be required to arrange payment for your article.

Please note that *Nature Ecology & Evolution* is a Transformative Journal (TJ). Authors may publish their research with us through the traditional subscription access route or make their paper immediately open access through payment of an article-processing charge (APC). Authors will not be required to make a final decision about access to their article until it has been accepted. [Find out more about Transformative Journals](https://www.springernature.com/gp/open-research/transformative-journals)

Authors may need to take specific actions to achieve [compliance with funder and institutional open access mandates](https://www.springernature.com/gp/open-research/funding/policy-compliance-faqs). If your research is supported by a funder that requires immediate open access (e.g. according to [a](https://www.springernature.com/gp/open-research/funding/policy-compliance-faqs)

21[Plan S principles](https://www.springernature.com/gp/open-research/plan-s-compliance)) then you should select the gold OA route, and we will direct you to the compliant route where possible. For authors selecting the subscription publication route, the journal's standard licensing terms will need to be accepted, including [a href="https://www.nature.com/nature-portfolio/editorial-policies/self-archiving-and-license-to-publish"](https://www.nature.com/nature-portfolio/editorial-policies/self-archiving-and-license-to-publish). Those licensing terms will supersede any other terms that the author or any third party may assert apply to any version of the manuscript.

[REDACTED]

[REDACTED]

Reviewer #1:

Remarks to the Author:

I am happy with the responses and text changes the authors have done to address my comments, but I have a couple of minor remarks:

1. Apparently I missed that the growth rates were reported in the original manuscript. After seeing that there were in fact significant growth rate effects when expressing RamF at 30 and 37 degrees I think the sentence "Although RamF resulted in minimal fitness cost at 37 °C ..." (page 15) is still slightly misrepresenting. Even if a difference in growth rate may appear small, if it is detectable as a change in growth rate it is many orders of magnitude above the limit of selection in bacterial populations. So I don't think you can claim a "minimal fitness cost".

2. I agree with reviewer 2 that the effect of MazF is an activity rather than a function. In my mind, "activity" refers to a biochemical/biophysical process (enzymatic activity, binding activity, inhibitory activity, etc) while "function" refers to a biologically significant activity (i.e. the activity that has evolved and is maintained by selection). For example, enzymes may have promiscuous activities (e.g. acting on non-native or non-natural substrates, or catalysing alternative reactions with their native substrate). If such activities are not subject to purifying or positive selection it does not make sense to call them "functions", but they are still significant and measurable activities that could come under selection and allow the enzyme to evolve a new function. Does this make sense?

Reviewer #2:

Remarks to the Author:

I have really enjoyed seeing the additional experiments in the revised version of the manuscript as well as the edits done, mainly in the discussion. I appreciate toning down the claims about the fitness cost of random proteins, relating to previous studies. I value the debate about why only one hit from the tested sequence space affected the MazF levels and how this relates to de novo gene emergence in vivo. I think that all these aspects really improved the manuscript significantly.

While I really appreciate this study, I have to say that I still hold my reservations to the specificity of this effect, which seems to be very sensitive to the actual expression levels of RamF. The initial version of the manuscript already showed that the effect is reverted by chaperone overexpression and the additional experiments (Figure S5F) imply that RamF levels may also be affected throughout the experiments. For now, we know that RamF levels are quite significantly affected by the Lon protease (the only one tested) which makes me wonder whether some effect would also be observed if tested in other protease deletion strains. The differences in MazF levels could then easily be combinations of such effects. This also makes me ask whether the authors have considered this in light of the last paragraph starting on bottom of page 7 (starting by "Because the activity of RamF depends...").

In general, the expression levels of RamF have not been debated while it can have significant effects on the observed phenomena (I think that this issue was also raised by another reviewer). The expression levels of de novo emerging genes are typically very low, and it seems that the phenomenon observed here is quite dependent on the RamF levels. Do the authors consider such effects possible by de novo emerging genes?

Given the number of experiments that have already been done here, I understand the reluctance of the authors to perform additional experiments with the IP-MS. Perhaps only follow-up studies will show how specific/unique the RamF example is. I would argue that exactly because there is so far no in vivo evidence of random protein interaction with chaperones, it is an important experiment to be done (especially since the authors reference a study which implied this interaction in vitro). It would also be great to see some follow up mechanistic/structural studies of the RamF-chaperone interactions (beyond Alpha-fold prediction).

Finally, I believe that the legend of Fig5 F vs G is swapped.

Reviewer #3:

Remarks to the Author:

The authors have made a commendable effort to carefully address all of the reviewers' comment. I think this study is an important contribution.

Author Rebuttal, first revision:

23We are delighted that our work is now accepted in principle. Below we respond to each new comment, indicating how the text has been modified. Reviewer comments are in black and our responses follow in blue.

Reviewer #1 (Remarks to the Author):

I am happy with the responses and text changes the authors have done to address my comments, but I have a couple of minor remarks:

1. Apparently I missed that the growth rates were reported in the original manuscript. After seeing that there were in fact significant growth rate effects when expressing RamF at 30 and 37 degrees I think the sentence "Although RamF resulted in minimal fitness cost at 37 °C ..." (page 15) is still slightly misrepresenting. Even if a difference in growth rate may appear small, if it is detectable as a change in growth rate it is many orders of magnitude above the limit of selection in bacterial populations. So I don't think you can claim a "minimal fitness cost".

Following this comment by Dr. Näsval, we edited the relevant sentence on page 15 (now page 14) to state that RamF production cost is most substantial at 30 °C, lower at 37 °C, and undetectable at 44 °C.

2. I agree with reviewer 2 that the effect of MazF is an activity rather than a function. In my mind, "activity" refers to a biochemical/biophysical process (enzymatic activity, binding activity, inhibitory activity, etc) while "function" refers to a biologically significant activity (i.e. the activity that has evolved and is maintained by selection). For example, enzymes may have promiscuous activities (e.g. acting on non-native or non-natural substrates, or catalysing alternative reactions with their native substrate). If such activities are not subject to purifying or positive selection it does not make sense to call them "functions", but they are still significant and measurable activities that could come under selection and allow the enzyme to evolve a new function. Does this make sense?

We respectfully maintain our position that RamF functions as a MazF inhibitor and that is a protein function that could be selected in nature. Moreover, we are concerned that the word 'activity' will imply *enzymatic* activity to many readers leading to significant confusion. We have, therefore, kept our current descriptor.

Reviewer #2 (Remarks to the Author):

I have really enjoyed seeing the additional experiments in the revised version of the manuscript as well as the edits done, mainly in the discussion. I appreciate toning down the claims about the fitness cost of random proteins, relating to previous studies. I value the debate about why only one hit from the tested sequence space affected the MazF levels and how this relates to de novo gene emergence in vivo. I think that all these aspects really improved the manuscript significantly.

While I really appreciate this study, I have to say that I still hold my reservations to the specificity of this effect, which seems to be very sensitive to the actual expression levels of RamF. The initial version of the manuscript already showed that the effect is reverted by chaperone overexpression and the additional experiments (Figure S5F) imply that RamF levels may also be affected throughout the experiments. For now, we know that RamF levels are quite significantly affected by the Lon protease (the only one tested) which makes me wonder whether some effect would also be observed if tested in other protease deletion strains. The differences in MazF levels could then easily be combinations of such effects. This also makes me ask whether the authors have considered this in light of the last paragraph starting on bottom of page 7 (starting by "Because the activity of RamF depends...").

In general, the expression levels of RamF have not been debated while it can have significant effects on the observed phenomena (I think that this issue was also raised by another reviewer). The expression levels of *de novo* emerging genes are typically very low, and it seems that the phenomenon observed here is quite dependent on the RamF levels. Do the authors consider such effects possible by *de novo* emerging genes?

Following these suggestions, we added a new paragraph at the end of page 13 discussing the possibility that RamF specificity might depend on its expression levels and that *de novo* genes must cross a certain expression threshold to be functional.

Given the number of experiments that have already been done here, I understand the reluctance of the authors to perform additional experiments with the IP-MS. Perhaps only follow-up studies will show how specific/unique the RamF example is. I would argue that exactly because there is so far no *in vivo* evidence of random protein interaction with chaperones, it is an important experiment to be done (especially since the authors reference a study which implied this interaction *in vitro*). It would also be great to see some follow up mechanistic/structural studies of the RamF-chaperone interactions (beyond Alpha-fold prediction).

We agree with the reviewer that there is merit in these experiments and may include them in our follow-up studies.

Finally, I believe that the legend of Fig5 F vs G is swapped.

We thank the reviewer for pointing out this small error, which is now fixed.

Reviewer #3 (Remarks to the Author):

The authors have made a commendable effort to carefully address all of the reviewers' comments. I think this study is an important contribution.

Thank you!

Final Decision Letter:

12th September 2023

Dear Professor Laub,

We are pleased to inform you that your Article entitled "Selection of a *de novo* gene that can promote survival of *E. coli* by modulating protein homeostasis pathways", has now been accepted for publication in Nature Ecology & Evolution.

Over the next few weeks, your paper will be copyedited to ensure that it conforms to Nature Ecology and Evolution style. Once your paper is typeset, you will receive an email with a link to choose the appropriate publishing options for your paper and our Author Services team will be in touch regarding any additional information that may be required

Due to the importance of these deadlines, we ask you please us know now whether you will be difficult to contact over the next month. If this is the case, we ask you provide us with the contact information (email, phone and fax) of someone who will be able to check the proofs on your behalf, and who will be available to address any last-minute problems . Once your paper has been scheduled for online publication, the Nature press office will be in touch to confirm the details.

Acceptance of your manuscript is conditional on all authors' agreement with our publication policies (see www.nature.com/authors/policies/index.html). In particular your manuscript must not be published elsewhere and there must be no announcement of the work to any media outlet until the publication date (the day on which it is uploaded onto our web site).

Please note that *Nature Ecology & Evolution* is a Transformative Journal (TJ). Authors may publish their research with us through the traditional subscription access route or make their paper immediately open access through payment of an article-processing charge (APC). Authors will not be required to make a final decision about access to their article until it has been accepted. [Find out more about Transformative Journals](https://www.springernature.com/gp/open-research/transformative-journals)

Authors may need to take specific actions to achieve [compliance with funder and institutional open access mandates](https://www.springernature.com/gp/open-research/funding/policy-compliance-faqs). If your research is supported by a funder that requires immediate open access (e.g. according to [Plan S principles](https://www.springernature.com/gp/open-research/plan-s-compliance))

26then you should select the gold OA route, and we will direct you to the compliant route where possible. For authors selecting the subscription publication route, the journal's standard licensing terms will need to be accepted, including <https://www.nature.com/nature-portfolio/editorial-policies/self-archiving-and-license-to-publish>. Those licensing terms will supersede any other terms that the author or any third party may assert apply to any version of the manuscript.

We welcome the submission of potential cover material (including a short caption of around 40 words) related to your manuscript; suggestions should be sent to Nature Ecology & Evolution as electronic files (the image should be 300 dpi at 210 x 297 mm in either TIFF or JPEG format). Please note that such pictures should be selected more for their aesthetic appeal than for their scientific content, and that colour images work better than black and white or grayscale images. Please do not try to design a cover with the Nature Ecology & Evolution logo etc., and please do not submit composites of images related to your work. I am sure you will understand that we cannot make any promise as to whether any of your suggestions might be selected for the cover of the journal.

You can generate the link yourself when you receive your article DOI by entering it here: <http://authors.springernature.com/share>.

[REDACTED]

P.S. Click on the following link if you would like to recommend Nature Ecology & Evolution to your librarian <http://www.nature.com/subscriptions/recommend.html#forms>

27** Visit the Springer Nature Editorial and Publishing website at http://editorial-jobs.springernature.com?utm_source=ejp_NEcoE_email&utm_medium=ejp_NEcoE_email&utm_campaign=ejp_NEcoE for more information about our career opportunities. If you have any questions please click [here](mailto:editorial.publishing.jobs@springernature.com).**